# miR-9 regulates basal ganglia-dependent developmental vocal learning and adult vocal performance in songbirds

**Zhimin Shi[1], Zoe Piccus[1], Xiaofang Zhang[1], Huidi Yang[1], Hannah Jarrell[1], Yan Ding[1], Zhaoqian Teng[1], Ofer Tchernichovski[2]\*, XiaoChing Li[1]\***

[1]Neuroscience Center of Excellence, Louisiana State University School of Medicine, New Orleans, United States; [2]Department of Psychology, Hunter College, New York, United States

**Abstract** miR-9 is an evolutionarily conserved miRNA that is abundantly expressed in Area X, a basal ganglia nucleus required for vocal learning in songbirds. Here, we report that overexpression of miR-9 in Area X of juvenile zebra finches impairs developmental vocal learning, resulting in a song with syllable omission, reduced similarity to the tutor song, and altered acoustic features. miR-9 overexpression in juveniles also leads to more variable song performance in adulthood, and abolishes social context-dependent modulation of song variability. We further show that these behavioral deficits are accompanied by downregulation of *FoxP1* and *FoxP2*, genes that are known to be associated with language impairments, as well as by disruption of dopamine signaling and widespread changes in the expression of genes that are important in circuit development and functions. These findings demonstrate a vital role for miR-9 in basal ganglia function and vocal communication, suggesting that dysregulation of miR-9 in humans may contribute to language impairments and related neurodevelopmental disorders.

DOI: https://doi.org/10.7554/eLife.29087.001

**\*For correspondence:**
tchernichovski@gmail.com (OT);
xli4@lsuhsc.edu (XCL)

**Competing interests:** The authors declare that no competing interests exist.

## Introduction

Like humans developing speech and language, juvenile songbirds learn to sing by imitating the songs of an adult tutor early in life (*Doupe and Kuhl, 1999*; *Immelmann and Hinde, 1969*; *Marler and Tamura, 1964*; *Tchernichovski et al., 2001*). Juvenile zebra finches (pupils) first hear and memorize a tutor's song, and then, at about post-hatching day 30 (30 d), the pupils begin to vocalize. Initially, juvenile songs are highly variable. Through thousands of trial-and-error practice sessions guided by auditory feedback, the juvenile song gradually matures into a stereotyped adult song that resembles the tutor's song (*Immelmann and Hinde, 1969*; *Konishi, 1965*; *Tchernichovski et al., 2001*). The adult song, however, exhibits a residual level of variability. Depending on the social context, adult males sing a courtship song directed toward a female (DS) or an undirected song (UDS) when singing alone (*Sossinka and Bohner, 1980*). These two types of songs exhibit subtle differences in acoustic features, and the DS is more stereotyped than the UDS (*Hessler and Doupe, 1999*; *Jarvis et al., 1998*; *Kao and Brainard, 2006*; *Murugan et al., 2013*).

The neural circuit that controls song behavior is organized into two pathways: a motor pathway and an anterior forebrain pathway (AFP) (*Figure 1A*). The motor pathway controls song production, while the AFP, a cortical-basal ganglia circuit required for song learning and maintenance, guides motor output through a reinforcement mechanism (*Bottjer et al., 1984*; *Brainard and Doupe, 2000*; *Doupe and Kuhl, 1999*; *Kao et al., 2005*; *Nottebohm et al., 1976, 1982*; *Olveczky et al., 2005*; *Simpson and Vicario, 1990*). Area X is a basal ganglia nucleus within the AFP. Lesions in Area X of juveniles prevent them from crystallizing their song (*Scharff and Nottebohm, 1991*). Area X

**eLife digest** When a cell needs to make a protein, it makes a temporary copy of the corresponding gene so that the genetic code can be carried to its protein-making machinery. When the temporary copy of the code is no longer needed, the cell destroys it. This system is fine-tuned by other small stretches of genetic code called microRNAs, which speed up the destruction and so help to switch genes off faster.

Two genes called *FOXP1* and *FOXP2* are known to have roles in speech and language development in humans. When these genes do not work properly, people have severe difficulties when speaking and understanding speech. But scientists know little about how the brain controls them. The brains of animals with backbones – like birds and mammals – make a microRNA called miR-9. Scientists thought miR-9 may control how active the *FOXP1* and *FOXP2* genes are in the brain.

Like humans, zebra finches communicate vocally. Young male birds learn to sing by imitating the song of an adult tutor, usually their father. The process is controlled by a brain region called "Area X". Now, Shi et al. report on the role of miR-9 in vocal learning and singing in zebra finches.

First, the gene for miR-9 was inserted into a virus-based genetic tool. Shi et al. then injected this virus into Area X of juvenile zebra finches, which delivered the gene to the brain cells and forced them to make excess miR-9. A control group received empty virus with no miR-9 gene for comparison. The juvenile finches then grew up with an adult bird that taught them to sing.

Shi et al. found that the birds that overproduced miR-9 did not learn as well as their normal counterparts. Their songs were shorter, they stuttered, and they missed out syllables, which meant that they simply sounded different to their tutors. These young birds also failed to change their tune in different situations, for example, when they met a female zebra finch. Examination of the birds' brains four weeks after the viral injection showed that the bird versions of the *FOXP1* and *FOXP2* genes were less active. There were also changes in other genes involved in brain circuit development.

Humans have a brain area like Area X, called the basal ganglia. The link between miR-9 and vocal learning provides a starting point to understand more about language in general. This could lead to improved understanding of conditions like stuttering, Tourette's syndrome, dyslexia and autism spectrum disorders.

DOI: https://doi.org/10.7554/eLife.29087.002

receives dopaminergic innervation from the midbrain ventral tegmental area (VTA), and it also modulates the activity of VTA dopaminergic neurons in response to auditory experience, implicating Area X in reinforcement learning (*Ding and Perkel, 2002*; *Gadagkar et al., 2016*; *Gale and Perkel, 2006*, *2010*; *Lewis et al., 1981*).

MicroRNAs (miRNAs) are small, non-protein-coding RNA molecules that regulate gene expression post-transcriptionally. miR-9 is an evolutionarily conserved miRNA that is highly expressed in vertebrate brains (*Landgraf et al., 2007*; *Luo et al., 2012*). In the zebra finch, miR-9 is expressed in Area X. Its expression is regulated during developmental vocal learning and in adult males singing undirected songs (*Shi et al., 2013*), suggesting that miR-9 plays an active role in these processes. miRNAs regulate gene expression by targeting the 3' untranslated regions (3'UTRs) of mRNAs, leading to mRNA degradation or suppression of protein synthesis. Many of the genes expressed in the nervous system have long 3'UTRs (*Mayr, 2016*), highlighting the importance of miRNAs in fine-tuning gene expression in nervous system development and function. FOXP1 and FOXP2 are a pair of paralogous transcription factors that have important roles in nervous system development. Deletions, mutations, and copy number variations of the *FOXP1* gene have been implicated in a range of neural developmental disorders, including language delay, intellectual disability, and autism spectrum disorders (ASDs) (*Hamdan et al., 2010*; *Horn et al., 2010*; *O'Roak et al., 2011*). Heterozygous mutations in the *FOXP2* gene cause severe speech and language impairments (*Lai et al., 2001*; *Vargha-Khadem et al., 1995*), accompanied by structural and functional abnormalities in multiple brain regions. These regions include the basal ganglia (*Watkins et al., 2002*), which is thought to be a component of the distributed neural circuitry that underlies speech and language (*Graham and Fisher, 2013*). FOXP1 and FOXP2 regulate the transcription of a

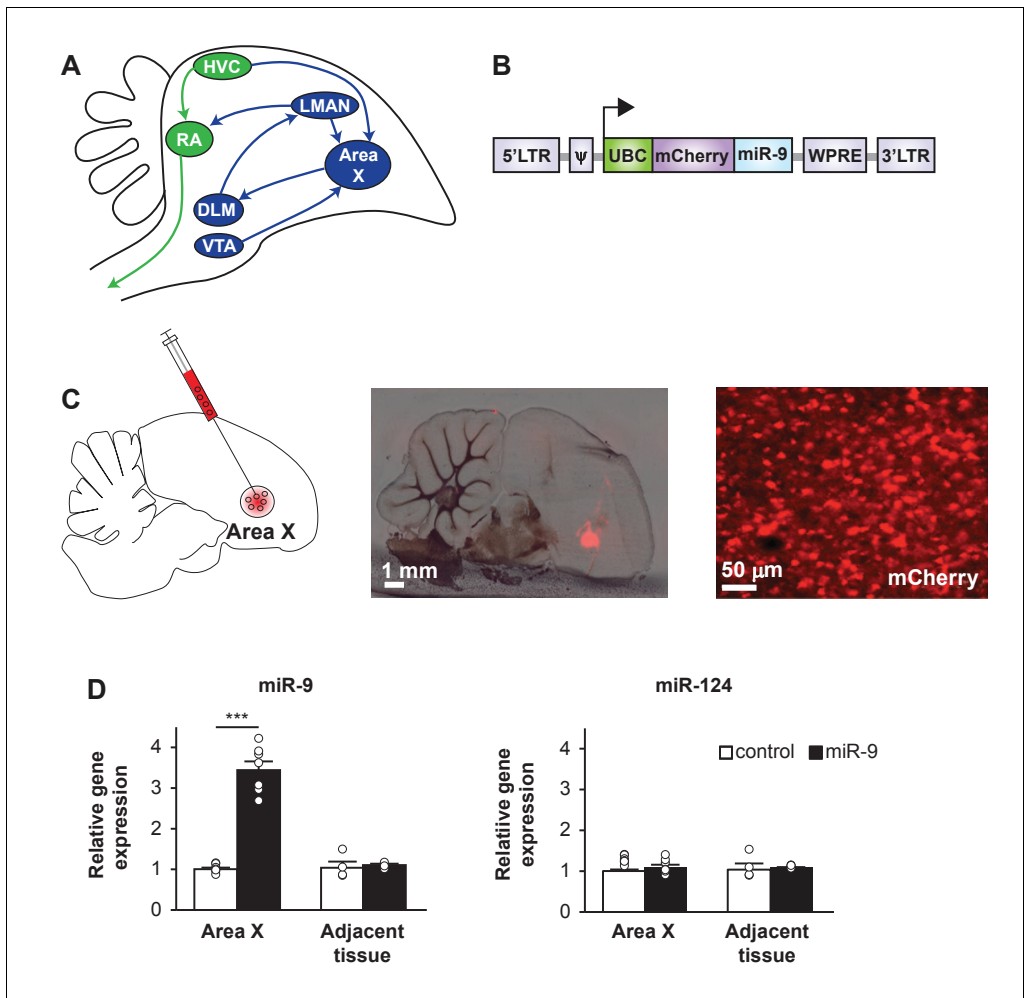

**Figure 1.** A lentiviral approach to manipulate miR-9 expression in Area X of the zebra finch brain. (**A**) Schematic drawing of the song control circuit in the zebra finch brain. The motor pathway (green), which connects HVC (used as a proper name) to RA (robust nucleus of the arcopallium) and eventually the vocal organ, controls song production. The anterior forebrain pathway (blue), which connects HVC to the basal ganglia nucleus Area X, DLM (medial nucleus of the dorsolateral thalamus), LMAN (lateral magnocellular nucleus), and then back to RA, is required for song learning. Area X also receives dopaminergic inputs from the VTA (ventral tegmental area). (**B**) The lentiviral vector used in this study expresses an mCherry fluorescent marker and miR-9 driven by the human ubiquitin promoter. (**C**) (Left) A diagram showing viral injection into Area X. (Middle and right) Sagittal sections of the zebra finch brain showing mCherry fluorescent signal in juvenile Area X four weeks after lentivirus injection. (**D**) The expression of miR-9 and miR-124 in Area X 4 weeks after injection with the lenti-miR-9 virus. p < 0.0001, t (12) = 11.21 for miR-9; p = 0.2879, t(12) = 1.112 for miR-124, unpaired t-test. n = 7 for Area X; n = 4 for adjacent tissue. Data are presented as mean ± SEM.

DOI: https://doi.org/10.7554/eLife.29087.003

The following figure supplements are available for figure 1:

**Figure supplement 1.** A lentiviral approach to manipulate miR-9 expression.

DOI: https://doi.org/10.7554/eLife.29087.004

**Figure supplement 2.** Impacts of viral injection on Area X volume and neuron number.

DOI: https://doi.org/10.7554/eLife.29087.005

large number of downstream genes, many of which are critically involved in neural differentiation, neurite outgrowth, synapse formation, and synaptic transmission (*Konopka et al., 2009*; *Spiteri et al., 2007*; *Tang et al., 2012*; *Vernes et al., 2007, 2011*). Thus, the functional dosage of FOXP1 and FOXP2 needs to be tightly regulated.

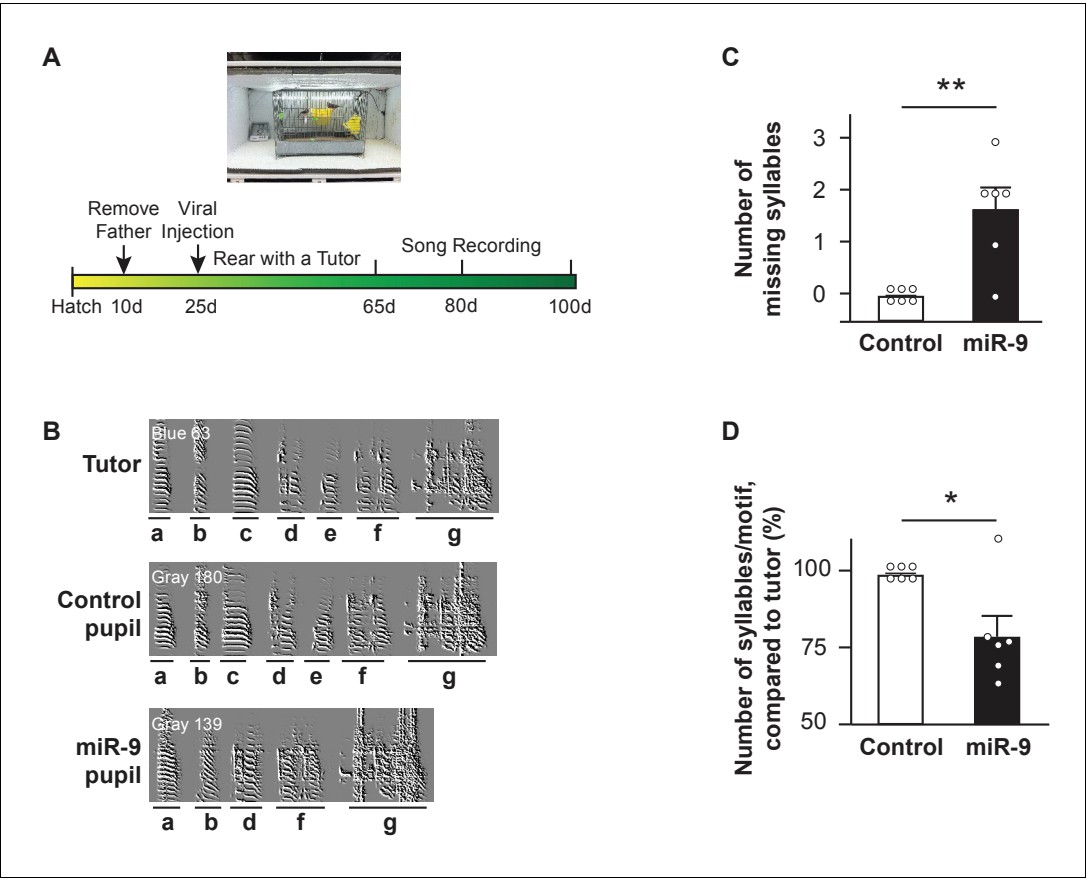

**Figure 2.** Syllable omission in pupils with miR-9 overexpression in juvenile Area X. (**A**) Experimental timeline for viral injection, song learning, and song recording. (**B**) Representative sonograms of a tutor, a control pupil, and an miR-9 pupil. Syllables are labeled alphabetically, and indicated with underlining below the sonograms. Note that the two pupils shared the same tutor, and syllables c and e were omitted from the miR-9 pupil's song. (**C**) Numbers of missing syllables in control and miR-9 pupils. p = 0.0075, U = 3, one-tailed Mann-Whitney U test; n = 6. (**D**) Numbers of syllables per motif in pupils' songs compared to their tutors' songs. p = 0.0325, U = 6, one-tailed Mann-Whitney U test; n = 6. Data are presented as mean ± SEM.

DOI: https://doi.org/10.7554/eLife.29087.006

The following source data is available for figure 2:

**Source data 1.**
DOI: https://doi.org/10.7554/eLife.29087.007

Recent in vitro studies have shown that miR-9 regulates the expression of *FOXP2* by targeting specific sequences in its 3'UTR (*Fu et al., 2014*; *Shi et al., 2013*). miR-9 also regulates avian *FoxP1* in embryonic chicken spinal cord (*Otaegi et al., 2011*). These findings raise the possibility that miR-9 has a role in language development through regulating *FOXP1* and/or *FOXP2*. Taking advantage of the unique vocal behavior and the underlying neural circuitry of songbirds, we sought to assess the consequences of miR-9 overexpression in Area X of juvenile zebra finches on vocal learning and performance. For these studies, we overexpressed miR-9 using a lentiviral approach. We report here that overexpression of miR-9 in juvenile Area X profoundly impairs basal ganglia-dependent developmental vocal learning in juveniles and impairs song performance in adulthood. We further show that these behavioral deficits are accompanied by downregulation of *FoxP1* and *FoxP2* expression, disruption of dopamine signaling, and widespread changes in the expression of numerous genes that are important for neural circuit development and function.

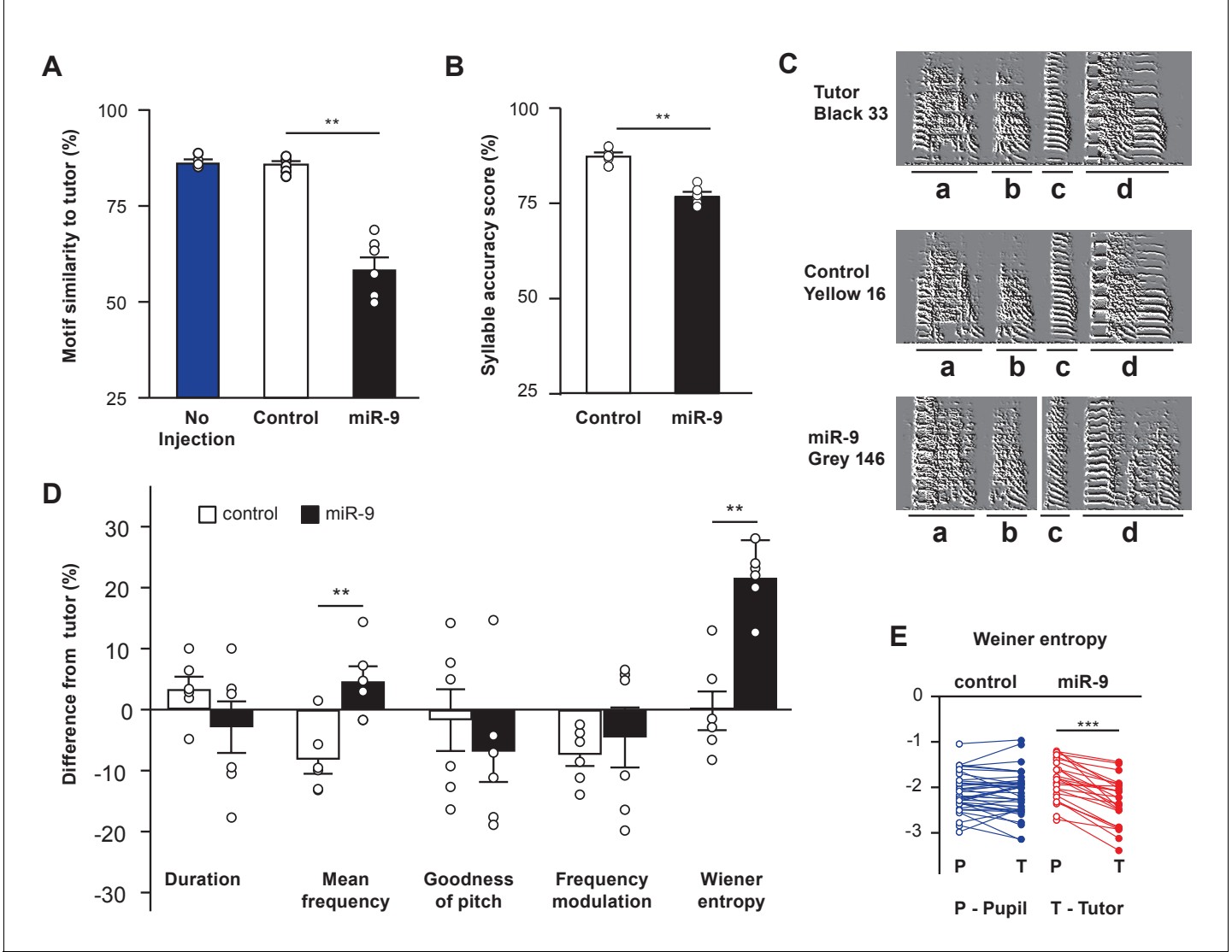

**Figure 3.** miR-9 overexpression in juvenile Area X impairs song learning. (**A**) Motif similarity scores of uninjected pupils, pupils injected with control virus, and pupils injected with miR-9 virus. p = 0.002, U = 0, two-tailed Mann-Whitney U Test; n = 6 for control and miR-9 pupils; n = 4 for uninjected pupils. (**B**) Syllable accuracy scores of control and miR-9 pupils. p = 0.002, U = 0, two-tailed Mann-Whitney U Test; n = 6. (**C**) Exemplar sonograms showing the fine structure of syllables of a tutor, a control pupil, and a miR-9 pupil. (**D**) Differences between syllable acoustic features of pupils' songs and tutors' songs. p = 0.01, U = 2 for mean frequency; p = 0.006, U = 1 for Wiener entropy, two-tailed Mann-Whitney U Test; n = 6. Data in (A), (B) and (D) are presented as mean ± SEM. (**E**) Syllable Wiener entropy of the control and miR-9 pupils compared to that of their tutors. p = 0.432, t(41) = 0.794 for control pupils; p < 0.001, t(24) = 8.245 for miR-9 pupils, paired t-test; control pupils: n = 42 syllables, 6 animals; miR-9 pupils: n = 25 syllables, 6 animals. Open circles represent pupils' syllables and filled circles represent tutors' syllables.
DOI: https://doi.org/10.7554/eLife.29087.008

The following source data and figure supplement are available for figure 3:

**Source data 1.**
DOI: https://doi.org/10.7554/eLife.29087.010
**Figure supplement 1.** Maximum motif similarity of control and miR-9 pupils at 100 d.
DOI: https://doi.org/10.7554/eLife.29087.009

## Results

### A lentiviral approach to manipulate miR-9 expression in Area X

The lentiviral vector that we used carried the mCherry fluorescent protein marker driven by the human ubiquitin promoter (hUBC) (*Edbauer et al., 2010*). We made a miR-9-expressing virus (lenti-miR-9) by inserting a miR-9 precursor sequence downstream of mCherry (*Figure 1B*). The control virus (lenti-control) carried mCherry alone. When tested in vitro, these lentiviruses effectively infected 293T cells, and overexpression of miR-9 downregulated the FOXP1 and FOXP2 proteins (*Figure 1—figure supplement 1A and B*). To test these viruses in vivo, we injected lenti-miR-9 or lenti-control virus into Area X of 25-day-old (25 d) male juvenile zebra finches, and examined miR-9 expression levels four weeks later using quantitative real-time PCR (qRT-PCR). In Area X injected with lenti-miR-9, miR-9 expression was increased more than three-fold compared to expression in Area X injected with lenti-control; the expression of an unrelated miRNA, miR-124, did not change (*Figure 1C and D*, $p < 0.0001$ for miR-9 and $p = 0.288$ for miR-124, n = 7). These results indicate that our lentiviral approach allowed effective overexpression of miR-9 in Area X in an miRNA-specific manner.

### miR-9 overexpression in juvenile Area X impairs vocal learning

We examined whether and how overexpression of miR-9 in juvenile Area X impairs vocal learning. In these experiments, 23–25 d male juvenile zebra finches (whose vocal learning was about to begin) were injected bilaterally into Area X with either the control virus (control pupils) or the lenti-miR-9 virus (miR-9 pupils). Each pupil was raised individually with an adult tutor from 30 d to 70 d. Pupils' songs were recorded at 65 d, 80 d, and 100 d (*Figure 2A*). A zebra finch song is made up of multiple renditions of a motif. A motif typically consists of 5–7 syllables rendered in a fixed sequence, with each syllable bearing distinct acoustic features (*Figure 2B*). We first analyzed pupils' songs recorded at 100 d (when they became adults). On the global motif structure level, control pupils imitated their tutors' song motif without syllable omission. By contrast, miR-9 pupils omitted some of their tutor's syllables and their song motifs were shorter than those of control pupils (*Figure 2B*). We quantified this phenomenon by manually counting the number of omitted syllables and the number of syllables per motif. We found that 5 of 6 miR-9 pupils omitted tutor syllables (*Figure 2C*, $p = 0.0075$; n = 6), and that the average number of syllables per motif was reduced by 24% in miR-9 pupils compared to control pupils (*Figure 2D*, $p = 0.0325$; n = 6).

We next examined how well the miR-9 pupils imitated the spectral structure and acoustic features of their tutors' songs using the Song Analysis Program software (SAP, [*Tchernichovski et al., 2000*]). SAP computes a similarity score, which reflects how similar two song motifs are, thus indicating how well a pupil learns its tutor's song. In quantifying motif similarity, we compared 20 pupil motif renditions to 10 tutor motif renditions, and averaged the 200 pairwise measurements for each animal. We found that miR-9 pupils exhibited a lower motif similarity score than control pupils; whereas the control pupils' motif similarity score was comparable to that of untreated pupils, indicating that virus injection alone did not affect song learning (*Figure 3A*, $p = 0.002$, two-tailed Mann-Whitney U Test; n = 6 for control and miR-9 pupils; n = 4 for untreated control pupils). We next ranked the 200 pairwise measurements, and averaged the 10 highest values to obtain a maximum similarity score for each pupil. The maximum similarity score of miR-9 pupils was significantly lower than that of control pupils (*Figure 3—figure supplement 1*, $p < 0.001$; n = 6), suggesting that even at their best performance, miR-9 pupils were not able to produce a good copy of their tutors' song.

To examine how well miR-9 pupils were able to imitate their tutors at the level of individual syllables, we quantified the syllable accuracy scores of control and miR-9 pupils. We found that miR-9 pupils imitated tutors' syllables less accurately than did control pupils (*Figure 3B and C*, $p = 0.002$, two-tailed Mann-Whitney U Test; n = 6). We also examined how individual acoustic features, including duration, mean frequency, goodness of pitch, frequency modulation, and Wiener entropy, differed between pupils and tutors. We found that the mean frequency and Wiener entropy differed between miR-9 pupils and tutors significantly more than between control pupils and tutors (*Figure 3D*, $p = 0.01$ for mean frequency; $p = 0.006$ for Wiener entropy, two-tailed Mann-Whitney U Test; n = 6). In addition, Wiener entropy differed significantly between miR-9 pupils and their tutors, but did not differ significantly between control pupils and their tutors ($p < 0.001$, $t(24) = 8.245$ for

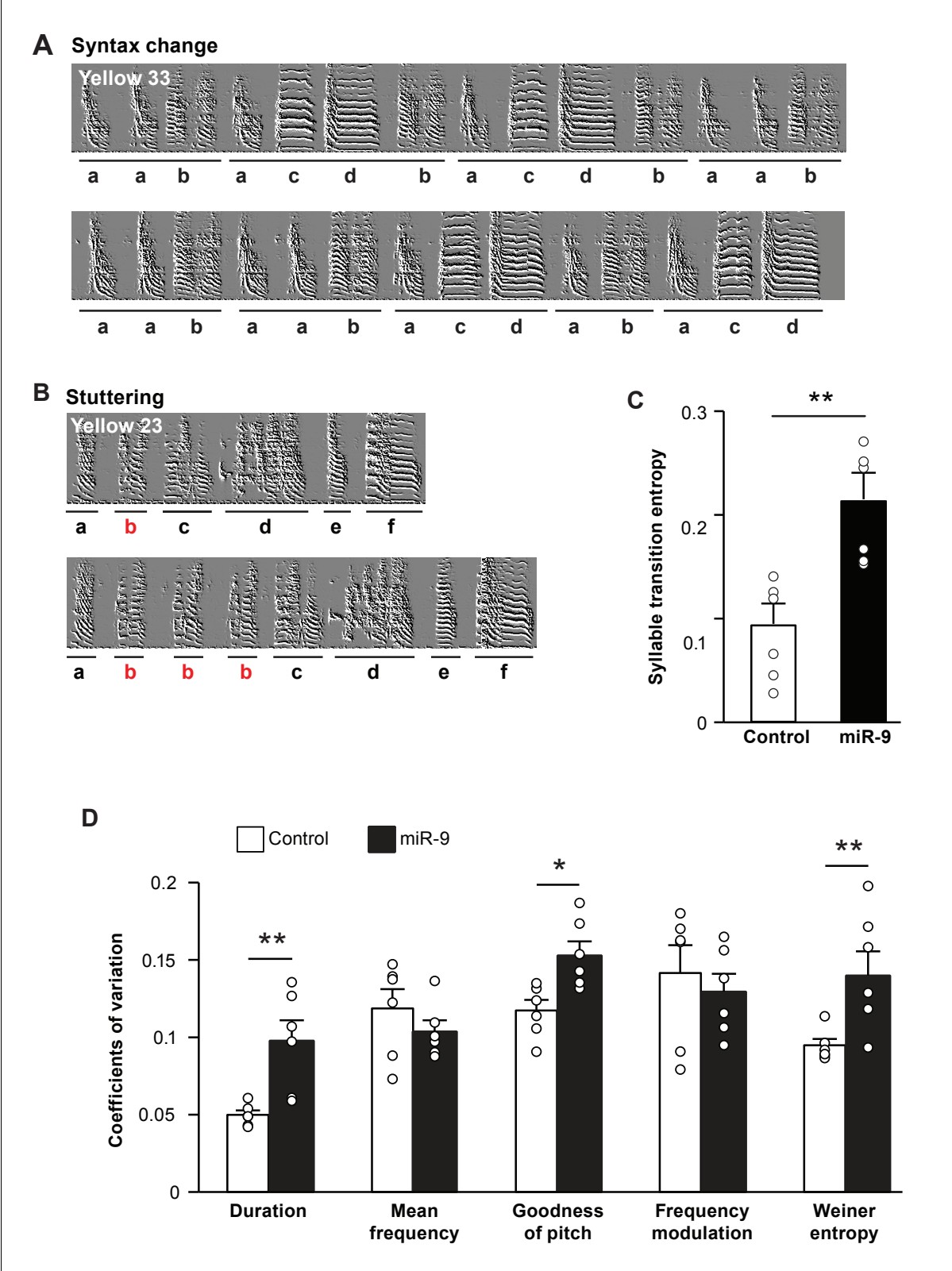

**Figure 4.** Overexpression of miR-9 in juvenile Area X impairs song performance in adulthood. (**A**) Representative sonograms of miR-9 pupil Yellow 33 showing an example of variation in syllable order. (**B**) Representative sonograms of miR-9 pupil Yellow 23 showing an example of syllable repetition. (**C**) Syllable transition entropy of adult control and miR-9 pupils who received viral injection as juveniles. p = 0.002, U = 0, two-tailed Mann-Whitney U test;
*Figure 4 continued on next page*

Figure 4 continued

n = 6. (D) Variations in syllable acoustic features of adult songs of control and miR-9 pupils. p = 0.009, U = 2 for duration; p = 0.015, U = 3 for goodness of pitch; and p = 0.009, U = 2 for Wiener entropy, two-tailed Mann-Whitney U test; n = 6. Data are presented as mean ± SEM.
DOI: https://doi.org/10.7554/eLife.29087.011

The following source data is available for figure 4:

**Source data 1.**
DOI: https://doi.org/10.7554/eLife.29087.012

miR-9 pupils; p = 0.432, t(41) = 0.794 for control pupils, paired t-test; control pupils: n = 42 syllable; miR-9 pupils: n = 25 syllables; n = 6 animals per group) (*Figure 3E*).

## miR-9 overexpression in juvenile Area X impairs song performance and abolishes social-context-dependent modulation of song variability in adulthood

To assess the effect of miR-9 overexpression in juvenile Area X on song performance in adulthood, we examined syllable sequence order in 100 d pupils' songs. A careful examination of sonograms showed that the songs of miR-9 pupils often exhibited switching of syllable order, truncation of motifs, and/or syllable stuttering (*Figure 4A and B*). We calculated syllable transition entropy to score these phenomena, where a higher transition entropy score reflects lower stereotypy of syllable sequences. We found that syllable transition entropy of miR-9 pupils was significantly higher than that of control pupils (*Figure 4C*, p = 0.002, two-tailed Mann-Whitney U Test; n = 6). We also measured trial-by-trial performance variation in syllable acoustic features across multiple renditions of songs of adult (100 d) miR-9 and control pupils. Among the acoustic features analyzed, duration, goodness of pitch, and Wiener entropy were significantly more variable in adult miR-9 pupils than in adult control pupils (*Figure 4D*, p = 0.009 for duration; p = 0.015 for goodness of pitch; and p = 0.009 for Wiener entropy, two-tailed Mann-Whitney U test; n = 6). These results indicate that overexpression of miR-9 in juvenile Area X leads to more variable song performance in adulthood.

It is known that the expression of miR-9 in Area X is upregulated by singing an undirected song (UDS) but not by singing a female-directed song (DS); furthermore, the acoustic features of UDS are more variable than those of DS (*Jarvis et al., 1998*; *Kao and Brainard, 2006*; *Leblois et al., 2010*; *Murugan et al., 2013*; *Shi et al., 2013*; *Teramitsu and White, 2006*). These findings prompted us to examine the possibility that miR-9 plays a role in modulating song variability according to social context. We recorded both UDS and DS of the same adult pupils (100 d), and examined the trial-by-trial variability in the constant fundamental frequency (cFF) of the same set of syllables in the two song types using a previously established method (*Kao and Brainard, 2006*; *Leblois et al., 2010*; *Murugan et al., 2013*). In control birds, the variability in syllable cFF was greater in UDS than in DS. In miR-9 pupils, however, the syllable cFF remained variable in DS, abolishing the social context-dependent modulation of syllable variability (*Figure 5A and B*, p = 0.006 for control pupils; p = 0.510 for miR-9 pupils, paired-t test; control pupils: n = 21 syllables, 6 animals; miR-9 pupils: n = 11 syllables, 6 animals). As juveniles are capable of producing an adult-like DS (*Kojima and Doupe, 2011*), we extended this analysis to 65 d and 85 d juveniles. Similar to the adults, the 65 d and 80 d control pupils produced a more stereotyped DS with reduced variability in cFF, whereas both 65 d and 80 d miR-9 pupils retained variability in DS (*Figure 5C*, p < 0.05 for 65 d and p < 0.01 for 80 d and 100 d groups, respectively; two-tailed Mann-Whitney U test; at 65 d, control pupils: n = 8 syllables, 3 animals; miR-9 pupils: n = 6 syllables, 4 animals; at 80 d, control pupils: n = 11 syllables, 5 animals; miR-9 pupils: n = 9 syllables, 5 animals; at 100 d, control pupils: n = 14 syllables, 6 animals; miR-9 pupils: n = 10 syllables, 6 animals). Together, these results suggest that miR-9 plays a role in modulating social-context-dependent song variability.

## The developmental process of vocal learning and performance

Vocal learning is a developmental process during which a highly variable juvenile song gradually matures into a stereotyped adult song that resembles the tutor's song. To gain insight into the role that miR-9 may play in this process, we tracked the developmental trajectory of song learning by analyzing songs of pupils produced at 65 d and 80 d. At both 65 d and 80 d, miR-9 pupils imitated

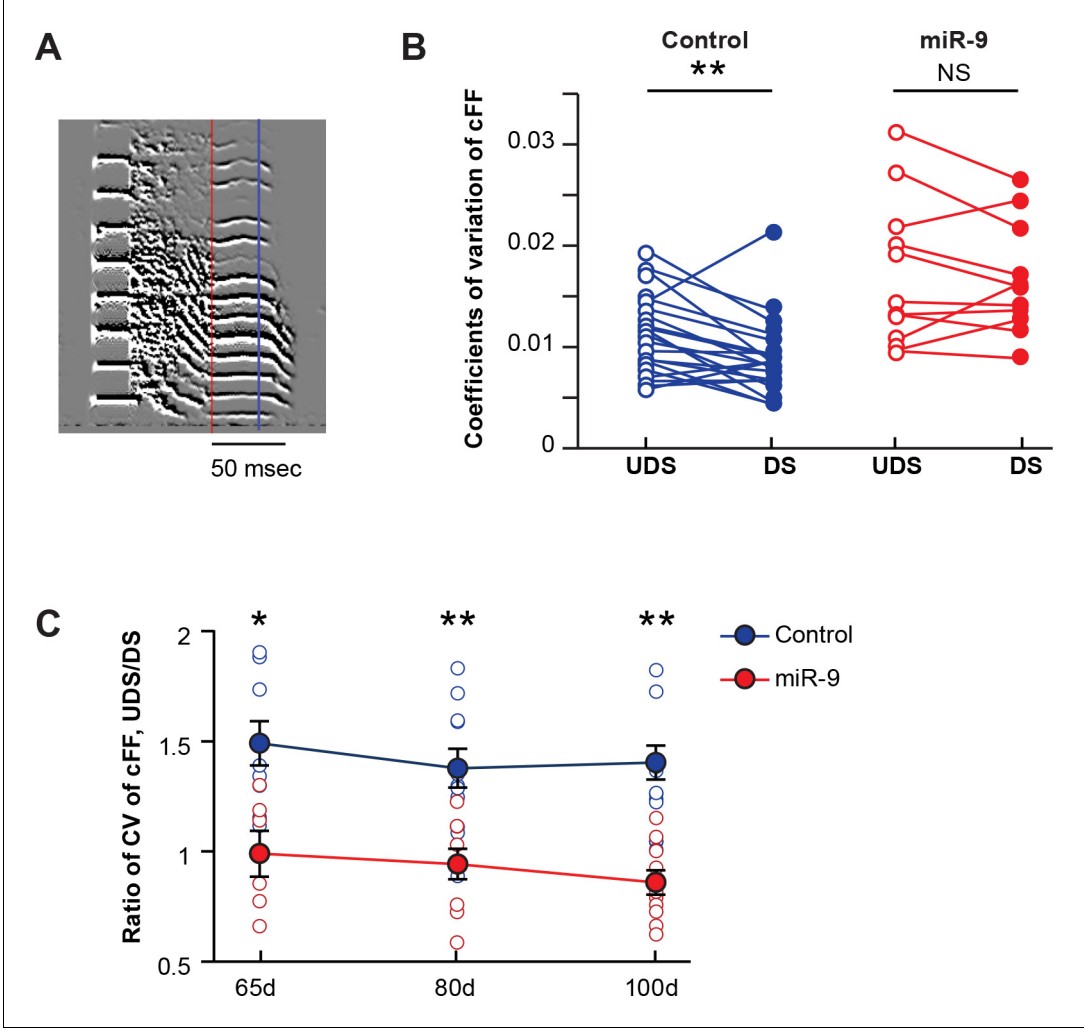

**Figure 5.** Overexpression of miR-9 in juvenile Area X abolishes social-context-dependent modulation of vocal performance. (**A**) An example syllable of a directed song. The lowest constant fundamental frequency (cFF) of a syllable segment between the red and the blue lines was measured with SAP. (**B**) Variations in cFF of UDS and DS of control and miR-9 pupils. Each pair of empty and filled circles connected by a line represents variations in cFF values measured for a syllable in the UDS and DS of an individual pupil. For control pupils, p = 0.006, t(20) = 3.062 (21 syllables, 6 birds); for miR-9 pupils, p = 0.510, t(10) = 0.684 (11 syllables, 6 birds), paired t-test. (**C**) Ratio of coefficients of variation of cFF of UDS/DS of control and miR-9 pupils during development. Plot compares (cFF variation of UDS)/(cFF variation of DS) between control and miR-9 pupils. Significance of the differences between control and miR-9 pupils: at 65 d, p = 0.0127, U = 5 (control pupils: n = 8 syllables, 3 birds; miR-9 pupils: n = 6 syllables, 4 birds); at 80 d, p = 0.0016, U = 10 (control pupils: n = 11 syllables, 5 birds; miR-9 pupils: n = 9 syllables, 5 birds); and at 100 d, p = 0.001, U = 10 (control pupils: n = 14 syllables, 6 animals; miR-9 pupils: n = 10 syllables, 6 animals), two-tailed Mann-Whitney U test. Data are presented as mean ± SEM.

DOI: https://doi.org/10.7554/eLife.29087.013

The following source data is available for figure 5:

**Source data 1.**

DOI: https://doi.org/10.7554/eLife.29087.014

poorly, and their songs were less similar to the tutors' song than those of control pupils (*Figure 6A*, p = 0.0022 for 65 d, 80 d and 100 d songs, two-tailed Mann-Whitney U test; n = 6 per group). We wondered whether miR-9 overexpression caused a developmental delay, causing miR-9 pupils to require a longer time to learn their song. To assess this, we extended motif similarity analysis to songs of 150 d pupils. We found that at 150 d, the similarity score of miR-9 pupils was significantly

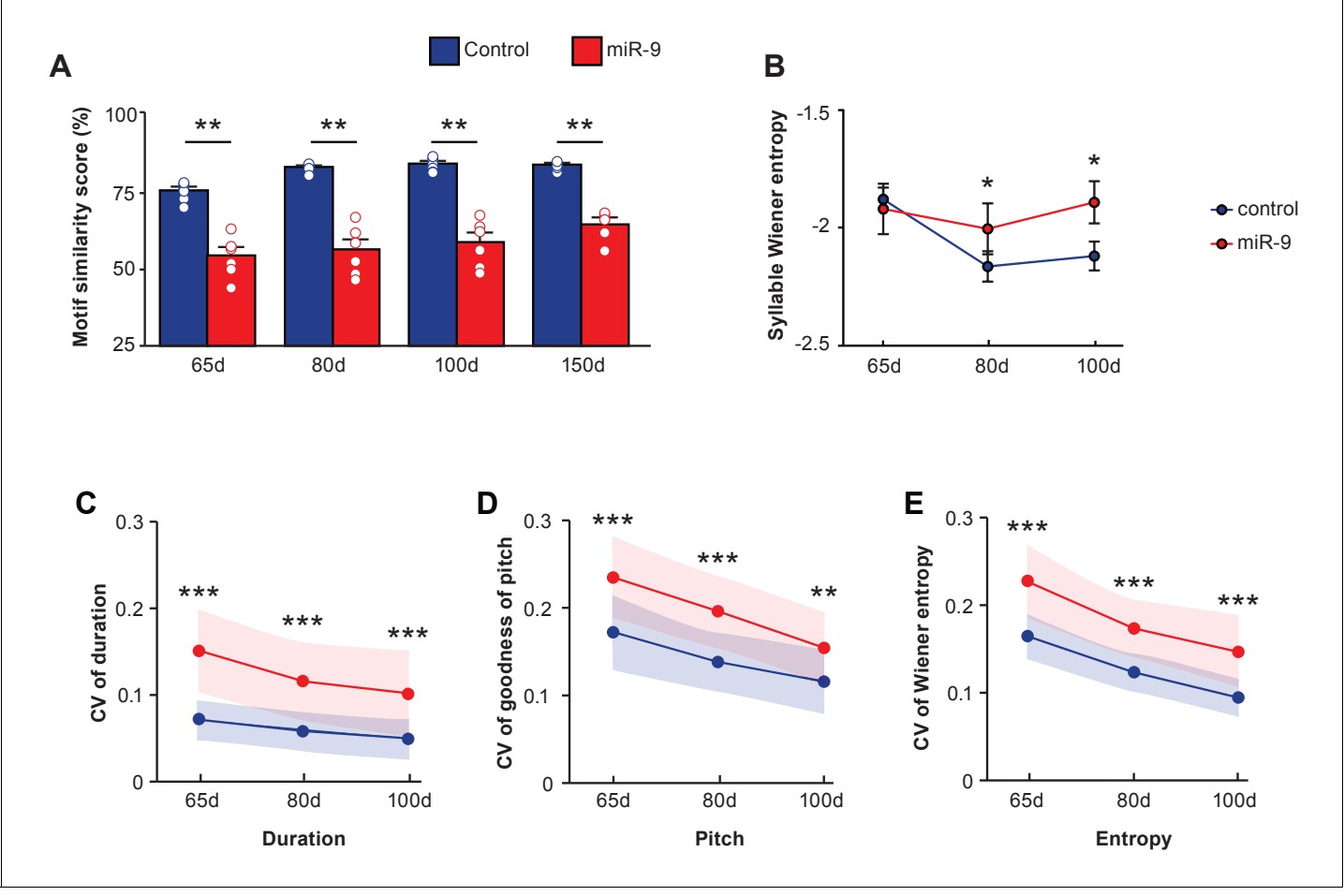

**Figure 6.** Developmental trajectory of vocal learning in virus-injected pupils. (**A**) Motif similarity scores of the control and miR-9 pupils during developmental learning. p = 0.002, U = 0 at 65 d, 80 d, and 100 d, and p = 0.008, U = 0 at 150 d, two-tailed Mann-Whitney U test; n = 6 for all groups. For song improvement from 65 d to 150 d: p < 0.001 for control pupils, and p = 0.137 for miR-9 pupils, one-way ANOVA; n = 6 for all groups. (**B**) Syllable Wiener entropy of control and miR-9 pupils during developmental learning. p = 0.482, U = 500 at 65 d; p = 0.028, U = 428 at 80 d; p = 0.020, U = 417 at 100 d, one-tailed Mann-Whitney U test. For Wiener entropy reduction: p = 0.002 for control pupils, p = 0.709 for miR-9 pupils, one-way ANOVA; n = 6 for all groups. (**C–E**) Variation in syllable performance with respect to duration (**C**), goodness of pitch (**D**), and Wiener entropy (**E**) during developmental song learning in control and miR-9 pupils. The dots represent mean coefficient of variation values for each acoustic feature at each age; shaded areas represent the range of standard deviations. For comparisons between the control and miR-9 pupils: p < 0.0001 for all acoustic features at all ages except p < 0.01 for pitch at 100 d, two-tailed Mann-Whitney U test. For variability reduction: p < 0.001 for duration, goodness of pitch, and Wiener entropy for both control and miR-9 pupils, one-way ANOVA. In (B–E), control pupils: n = 42 syllables, 6 animals; miR-9 pupils: n = 25 syllables, 6 animals at each age. Data are presented as mean ± SEM.

DOI: https://doi.org/10.7554/eLife.29087.015
The following figure supplement is available for figure 6:

**Figure supplement 1.** Amount of song production by control and miR-9 pupils at 65 d and 100 d.
DOI: https://doi.org/10.7554/eLife.29087.016

lower than that of control pupils (*Figure 6A*, p = 0.008, two-tailed Mann-Whitney U Test; n = 6). While the control pupils improved the similarity score of their song as they matured, miR-9 pupils did not improve their score from 65 d to 150 d (*Figure 6A*, p = 0 for control pupils and p = 0.137 for miR-9 pupils, one-way ANOVA; n = 6). We also measured syllable Wiener entropy changes during development. At 65 d, Wiener entropy of the control pupils and the miR-9 pupils was similar, but at both 80 d and 100 d, the Wiener entropy scores of miR-9 pupils were higher than those of the control pupils (*Figure 6B*, p = 0.482 for 65 d, p = 0.028 for 80 d, and p = 0.020 for 100 d, Mann-Whitney U test). The Wiener entropy of control pupils' syllables was reduced as they matured,

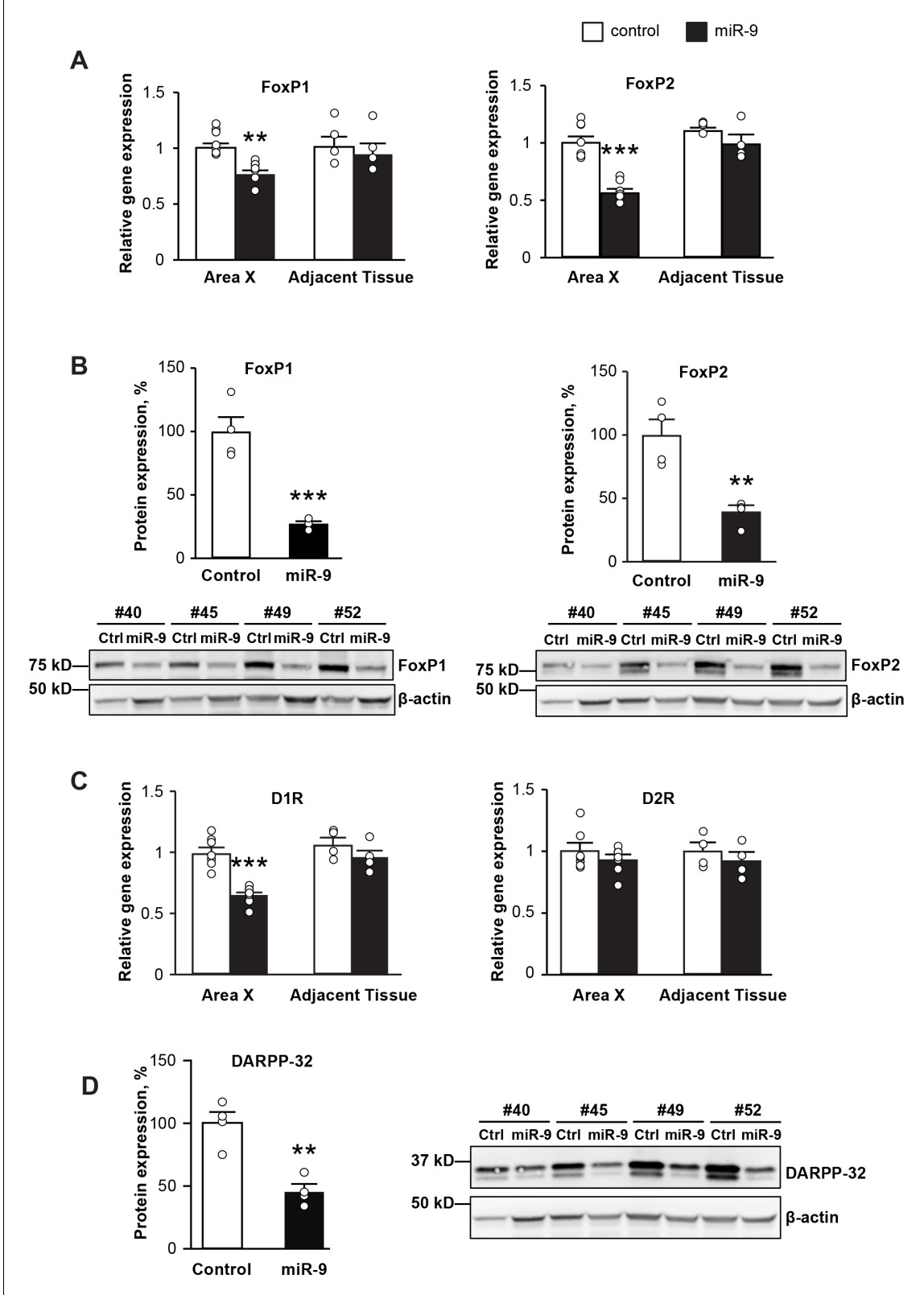

**Figure 7.** miR-9 overexpression in Area X downregulates the expression of *FoxP1* and *FoxP2* and disrupts dopamine signaling.  (A) Expression of *FoxP1* and *FoxP2* mRNAs in Area X measured 4 weeks after injection with the lenti-miR-9 virus by qRT-PCR. p = 0.006, t(12) = 4.608 for *FoxP1*; p < 0.0001, t(12) = 7.062 for *FoxP2*, unpaired t-test. n = 7 for Area X; n = 4 for adjacent tissue. (B) Western blot showing expression of FoxP1 and FoxP2 proteins in Area X 4 weeks after injection with the lenti-miR-9 virus. p = 0.0007, t(6) = 6.313 for FoxP1; p = 0.0037, t(6) = 4.608 for FoxP2, unpaired t-test; n = 4. (C)

*Figure 7 continued*

Expression of dopamine receptors D1R and D2R mRNAs in Area X 4 weeks after injection with the lenti-miR-9 virus measured by qRT-PCR. For D1R, p < 0.0001, t(12) = 6.441; for D2R, p = 0.3384, t(12) = 0.9971, unpaired t-test; n = 7 for Area X. (D) DARPP-32 protein expression in Area X 4 weeks after injection with lenti-miR-9 virus. p = 0.0022, t(6) = 5.104, unpaired t-test; n = 4. Data are presented as mean ± SEM. Lenti-control and lenti-miR-9 viruses were injected into Area X of contralateral hemispheres, #40, #45, #49 and #52 in western blot images are IDs of animals.
DOI: https://doi.org/10.7554/eLife.29087.017
The following figure supplement is available for figure 7:

**Figure supplement 1.** Western blots showing specificity of detection of FoxP2 and DARPP-32 proteins in Area X tissue indicated by inhibition of detection of (A) FoxP2 and (B) DARPP-32 after primary antibodies were pre-incubated with 10-fold molar excess of competing peptides.
DOI: https://doi.org/10.7554/eLife.29087.018

whereas the Wiener entropy of miR-9 pupils' syllables was not (*Figure 6B*, p = 0.002 for control pupils and p = 0.709 for miR-9 pupils, one-way ANOVA; n = 6 per group).

We also examined the trial-by-trial variability of syllable acoustic features including the duration, goodness of pitch, and Wiener entropy of songs produced during development. Syllable acoustic variability was reduced in songs of both control pupils and miR-9 pupils from 65 d to 100 d (*Figure 6C–E*, p < 0.001 for duration, pitch, and Wiener entropy for both control and miR-9 pupils [except p < 0.01 for duration for miR-9 pupils]; one-way ANOVA; control pupils: n = 42 syllables, 6 animals; miR-9 pupils: n = 25 syllables, 6 animals). However, variability in each of these acoustic features was significantly higher in miR-9 pupils than in control pupils at all ages (*Figure 6C–E*, p < 0.0001 for duration, pitch, and Wiener entropy for each age, two-tailed Mann-Whitney U test; control pupils: n = 42 syllables, 6 animals; miR-9 pupils: n = 25 syllables, 6 animals). These data indicate that miR-9 overexpression in juvenile Area X leads to higher variability in songs throughout maturation. This high level of trial-by-trial variation in acoustic features of miR-9 pupils may contribute to their inability to imitate the tutor's song accurately.

We wondered whether miR-9 overexpression affected the amount of song production, which subsequently contributed to impaired song learning. To assess this possibility, we examined the amount of singing by pupils at 65 d and 100 d. We found that at both 65 d and 100 d, miR-9 pupils sang slightly more syllables than control pupils, but the differences were not significant (*Figure 6—figure supplement 1*, p = 0.419 for 65 d; p = 0.109 for 100 d, two-tailed Mann-Whitney U Test; n = 6). Thus, it is unlikely that the amount of singing contributed to the effect of miR-9 on song learning.

## Impairments in vocal learning and performance are accompanied by FoxP1 and FoxP2 downregulation, interrupted dopamine signaling, and widespread changes in the expression of genes important for neural circuit development and function

To understand the molecular substrates underlying the impairments in vocal learning and performance described above, we examined changes in gene expression in Area X upon miR-9 overexpression. We first examined *FoxP1* and *FoxP2* mRNA and protein expression in Area X of juveniles four weeks after viral injection. We found that the expression levels of both *FoxP1* and *FoxP2* mRNAs were significantly downregulated in Area X injected with lenti-miR-9 compared to Area X injected with lenti-control (*Figure 7A*, p = 0.006 for *FoxP1* and p < 0.0001 for *FoxP2*, n = 7), whereas no change in expression of either *FoxP1* or *FoxP2* mRNA was found in tissue adjacent to Area X. We found that both FoxP1 and FoxP2 protein levels were also downregulated in Area X injected with the lenti-miR-9 virus compared to controls (*Figure 7B*, p = 0.0007 for FoxP1 and p = 0.0037 for FoxP2, n = 4).

The neurotransmitter dopamine plays an important role in modulating basal ganglia circuit plasticity and song stereotypy (*Ding and Perkel, 2002*; *Leblois et al., 2010*; *Murugan et al., 2013*; *Sasaki et al., 2006*). Both the dopamine D1 (D1R) and D2 (D2R) receptors are expressed in Area X (*Kubikova et al., 2010*), and D1R is regulated by FoxP2 (*Murugan et al., 2013*). Therefore, we examined the expression levels of D1R and D2R in Area X in which miR-9 was overexpressed. We found that four weeks after viral injection, D1R was downregulated in Area X injected with the lenti-miR-9 virus compared to Area X injected with the lenti-control virus, whereas the expression of D2R was unchanged (*Figure 7C*, p < 0.0001 for D1R and p = 0.3384 for D2R, unpaired t-test; n = 7). DARPP-32, a 32 kDa dopamine- and cAMP-regulated phosphoprotein, is a major signal transduction

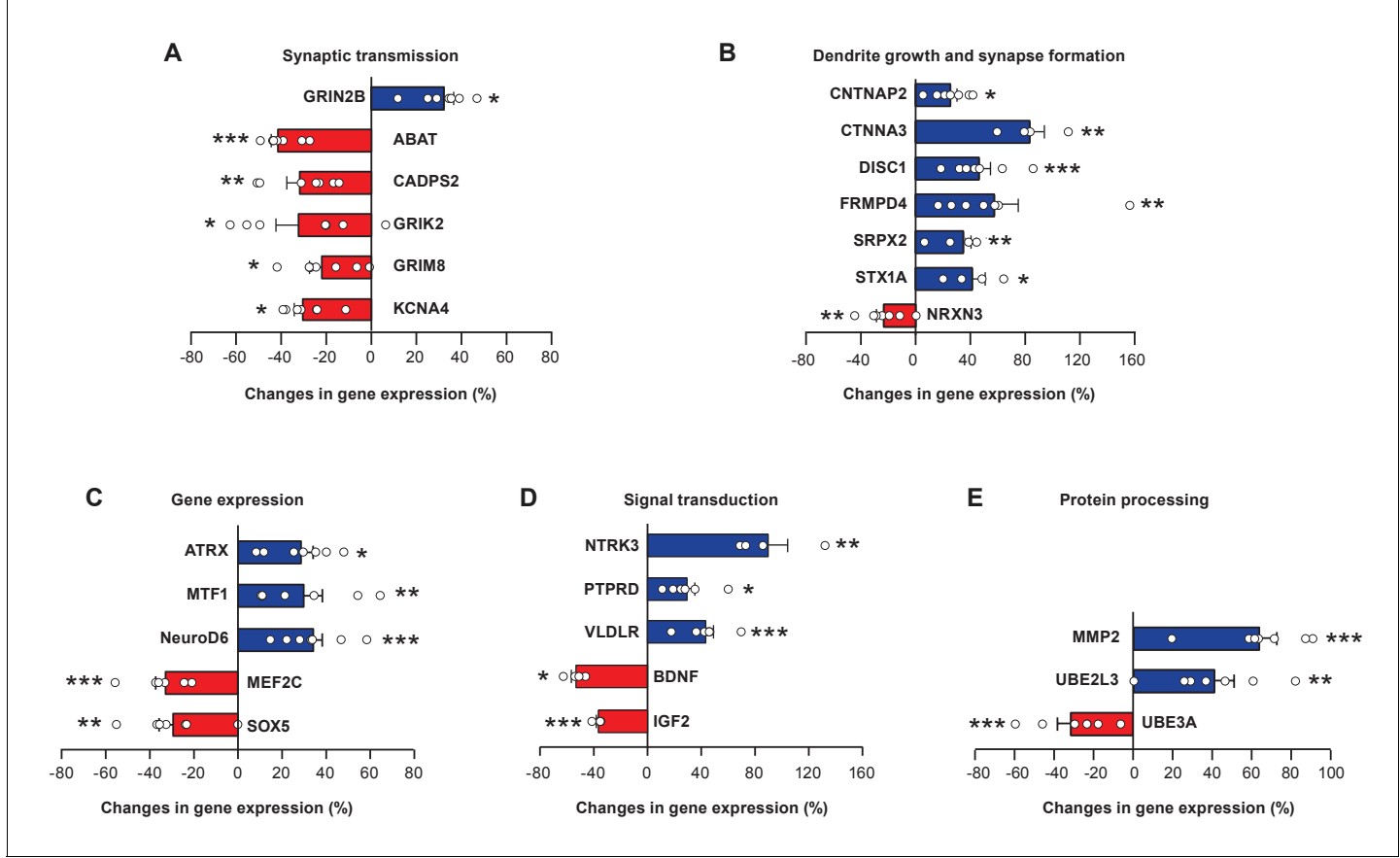

**Figure 8.** miR-9 overexpression alters expression levels of numerous FoxP1 and FoxP2 downstream target genes in Area X. Blue bars, upregulated genes; red bars, downregulated genes. Genes are grouped into five functional groups: (A) neurotransmitter receptors and channels; (B) proteins involved in dendritic growth and synapse formation; (C) transcription factors; (D) signaling molecules; and (E) proteins for protein processing and degradation. Reverse transcription and qRT-PCR were performed twice, and qRT-PCR was performed in triplicate. *p < 0.05, **p < 0.01, ***p < 0.001, unpaired t-test; n = 7 for all genes, except for *BDNF, CTNNA3, NRTK,* and *STX1A*, n = 4. Data are presented as mean ± SEM.
DOI: https://doi.org/10.7554/eLife.29087.019

The following figure supplement is available for figure 8:

**Figure supplement 1.** Downstream target genes of FoxP1 and FoxP2 that were unaffected by miR-9 overexpression in Area X.
DOI: https://doi.org/10.7554/eLife.29087.020

component acting downstream of dopamine receptors, and is highly expressed in striatal medium spiny neurons (*Greengard et al., 1999*; *Murugan et al., 2013*). We found that DARPP-32 protein level was significantly downregulated in Area X injected with the lenti-miR-9 virus compared to Area X injected with the lenti-control virus (*Figure 7D*, p = 0.0022, unpaired t-test; n = 4).

FOXP1 and FOXP2 regulate a large number of downstream transcriptional target genes that have important roles in neural circuit development and functions (*Konopka et al., 2009*; *Spiteri et al., 2007*; *Vernes et al., 2011*). We curated zebra finch homologs of 37 FOXP1 and/or FOXP2 down-stream genes based on published literature (*Bowers and Konopka, 2012*; *Konopka et al., 2009*; *Spiteri et al., 2007*; *Tang et al., 2012*), and examined whether their expression levels changed upon miR-9 overexpression. We found widespread changes in gene expression: 26 of 37 tested genes changed their expression in juvenile Area X 4 weeks after miR-9 viral injection. Consistent with the fact that FOXP1 and FOXP2 are capable of bi-directional regulation of gene expression, either enhancing or repressing gene expression (*Li et al., 2004*), we observed bi-directional changes in the expression of these downstream genes: 15 upregulated and 11 downregulated. The magnitudes of the changes, however, were moderate; a majority of these genes increased or decreased their expression by less than 50%. We grouped these 26 genes into functional modules: components of synaptic transmission (*ABAT, CADPS2, GRIN2B, GRIK2, GRM8, KCNA4*), cell adhesion molecules

important for dendrite growth and synapse formation (*CNTNAP2, CTNNA3, DISC1, FRMPD4, NRXN3, SRPX2, STX1A*), transcriptional regulators (ATRX , *MEF2C, MTF1, NeuroD6, SOX5*), signaling molecules (BDNF, *IGF2, NTRK3, PTPRD, VLVDR*), and peptidase and components of the ubiquitin protein degradation pathway (*MMP2, UBE2L3, UBE3A*) (*Figure 8A–E*, p < 0.05 for all genes, t-test; n = 7 for all genes except for *BDNF, CTNNA3, NTRK3*, and *STX1A*, n = 4). The symbols, names, functions, and possible associations with various neural developmental disorders of these genes are summarized in *Supplementary file 1*. A fraction (11 of 37) of the FOXP1 and FOXP2 downstream genes that we tested did not, however, change in expression following miR-9 overexpression (*Figure 8—figure supplement 1*). Among the possible explanations are that the regulatory relationships between FOXP1 or FOXP2 and these genes may be species-specific, or that additional cellular regulatory mechanisms may have contributed to the gene expression levels that we observed.

## Discussion

We show here that overexpression of miR-9 in the basal ganglia nucleus Area X of juvenile zebra finches impaired developmental vocal learning and adult vocal performance. On the global motif structural level, the most pronounced impairment was that birds with miR-9 overexpression sang shorter song motifs, omitting some of the tutor syllables. This phenomenon in miR-9 pupils (in which *FoxP1* and *FoxP2* were downregulated) may mirror the limited vocabulary observed in human individuals carrying deletions in the *FOXP1* gene, who typically exhibit a working vocabulary of fewer than 100 words at the age of seven (*Horn et al., 2010*). Songs of birds with miR-9 overexpression in Area X exhibited higher trial-by-trial variability, which was reflected in a more variable sequence of syllable order, truncated motifs, and syllable stuttering. Syntax change has not been reported in prior studies in which *FoxP2* was either knocked down or overexpressed in Area X (*Haesler et al., 2007*; *Heston and White, 2015*; *Murugan et al., 2013*). That miR-9 regulates the expression of both *FoxP1* and *FoxP2* and potentially of other genes in Area X may explain the robust deficits in vocal learning we observed here. Effects of miR-9 on song performance also occurred in a social context-dependent manner. Similar to birds with reduced *FoxP2* expression in adult Area X (*Murugan et al., 2013*), birds with miR-9 overexpression failed to modulate song variability when singing a directed song. Perhaps not coincidentally, miR-9 expression is higher in Area X when adult males naturally sing an undirected song (*Shi et al., 2013*). It appears that, whether naturally or artificially induced, higher miR-9 levels in Area X render the circuit permissive to a more variable song or interfere with the production of a more stereotyped directed song.

Vocal learning is a developmental process during which a less-structured and highly variable juvenile song gradually transitions to a stereotyped adult song that matches the tutor's song (*Immelmann and Hinde, 1969*; *Tchernichovski et al., 2001*). The effects of miR-9 overexpression on song imitation were apparent at 65 days after hatching, and these birds failed to improve their imitation thereafter (*Figure 6A*). Acoustic features of syllables of miR-9 pupils also exhibited higher trial-by-trial variability throughout development (*Figure 6C–E*). It is not clear whether or how higher song variability affects a pupil's ability to match its song to a tutor's song. In normal birds, the developmental vocal learning process is accompanied with a gradual increase in the miR-9 level in Area X, reaching its high point as juveniles become adults, and songs become stabilized; meanwhile, throughout the process, the FoxP2 level gradually decreases (*Haesler et al., 2004*; *Shi et al., 2013*; *Teramitsu et al., 2004*). Our current data indicate that an artificially increased miR-9 level (and reduced FoxP2 level) does not result in premature stabilization of a juvenile's song nor in an increase in similarity to the tutor's song. Rather, miR-9 overexpression appears to lock Area X in a plasticity-dominant state, preventing further progression toward song stereotypy. Previous lesion studies have shown that lesions in Area X and lMAN, two nuclei in the anterior forebrain pathway, have different consequences on song development. While lesions in lMAN lead to a prematurely stabilized song, birds with Area X lesions never achieve song stability as control birds do (*Scharff and Nottebohm, 1991*). Although the experimental approaches used in these studies are unrelated (molecular manipulation vs. electrolytic lesion), our observations are more aligned with the effects of lesions in Area X, suggesting that the effect of miR-9 overexpression (gene manipulation) might be restrained by circuit functions, in this case functions of Area X.

According to the reinforcement learning theory, Area X and the AFP guide song motor learning through processes that evaluate the motor output patterns according to auditory feedback, and reinforce favorable motor actions. These processes involve dopamine signaling via D1R and D2R receptors expressed in Area X (*Kubikova et al., 2010*) and dopaminergic projections from the VTA to Area X (*Ding and Perkel, 2002*; *Doupe et al., 2005*; *Doya and Tesauro, 1995*; *Gadagkar et al., 2016*; *Lewis et al., 1981*). Dopamine levels in Area X are higher when birds sing DS (*Sasaki et al., 2006*). Blocking D1R pharmacologically or reducing D1R expression by knocking down FoxP2 in Area X results in a more variable DS (*Leblois et al., 2010*; *Murugan et al., 2013*). Our findings that miR-9 overexpression in Area X selectively downregulated D1R but not D2R provide further evidence of the importance of dopamine signaling in vocal learning and performance. Studies in mammals suggest that the D1R-expressing direct pathway and the D2R-expressing indirect pathway act in an opposing but coordinated manner in the basal ganglia to finely control the timing and synchronization of motor actions; imbalances between the two pathways may lead to movement and cognitive disorders (*Cazorla et al., 2015*; *Gerfen and Surmeier, 2011*). The downregulation of D1R but not D2R in Area X, as a consequence of miR-9 overexpression, may have produced a functional imbalance between the two signaling pathways, which may have contributed to impaired song learning and performance. In normal birds, miR-9 expression in Area X is regulated during vocal development, and is upregulated by adult singing UDS (*Shi et al., 2013*). Thus, the miR-9–FoxP2–dopamine signaling network provides a mechanism that dynamically adjusts the functional balance between the D1R and D2R signaling pathways, allowing birds to adapt to the changing maturation state of song in juveniles and to modulate song stereotypy according to the social context in which a song is produced.

Many of the FOXP1 and FOXP2 downstream transcriptional target genes were identified by genome-wide chromatin immunoprecipitation assays, which depend on the binding of transcription factors FOXP1 or FOXP2 to promoter sequences (*Konopka et al., 2009*; *Spiteri et al., 2007*; *Vernes et al., 2011*). The changes in expression of these downstream genes in Area X when *FoxP1* and *FoxP2* were downregulated by miR-9 provides further evidence for a regulatory relationship between these genes and FOXP1 or FOXP2 in the basal ganglia circuit critical for vocal communication. Among the genes we tested, many have direct roles in neural circuit development and functions, and their dysregulation has been implicated in various neural developmental disorders. For example, *GRIK2*, *GRIN2B*, and *GRM8* encode subunits of glutamate receptors. Their altered expression can affect synaptic transmission of Area X neurons. *KCNA4* encodes a voltage-gated potassium channel (*Ovsepian et al., 2016*), and mutations in *KCNA4* have been identified in patients exhibiting linguistic disabilities, attention deficit hyperactivity disroder (ADHD), and cognitive impairments (*Kaya et al., 2016*). *MMP2,* a member of the matrix metalloproteinase family, plays critical roles in synaptogenesis, dendrite remodeling, and neurogenesis (*Fujioka et al., 2012*). In songbird brain, MMP2 has been implicated in angiogenesis and neurogenesis in HVC, another song nucleus essential for song learning and production (*Kim et al., 2008*), suggesting that MMP2 may have a role in Area X circuit development as well. *CNTNAP2* encodes a cell surface protein that belongs to the neurexin family, and is important for synaptic formation and clustering of K$^+$ channels at synaptic terminals (*Poliak et al., 1999*). FOXP2 is known to regulate *CNTNAP2* expression by binding to a regulatory sequence in its first intron, and *CNTNAP2* dysfunctions have been implicated in specific language impairments, intellectual disabilities, and ASDs (*Rodenas-Cuadrado et al., 2014*; *Vernes et al., 2008*). *DISC1* (*Disrupted in schizophrenia 1*) plays important roles in cell migration and dendrite development, and it has been linked to schizophrenia, bipolar disorder, depression, and ASDs (*Millar et al., 2000*; *Thomson et al., 2013*). The widespread changes in gene expression caused by miR-9 overexpression touch upon an array of cellular functions, including synaptic transmission, dendrite growth, synapse formation, gene regulation, neurotrophin (BDNF) signaling, and protein degradation. We only tested expression changes in a small fraction of FOXP1 and FOXP2 downstream genes; however, given the fact that two-thirds (26 of 37) of these genes changed expression, it is likely that many of the genes we did not test also changed their expression. Thus, a large number of these affected genes may have collectively contributed to the deficits in vocal behavior.

We noted that the magnitudes of changes in gene expression were moderate. Intriguingly, a recent study of a large cohort of schizophrenia patients found moderate changes (ranging from 10% to 40%) in the expression of several hundred genes in the prefrontal cortex (*Fromer et al., 2016*).

Our observations support the emerging view that subtle but broad changes in gene expression might be a molecular signature underlying complex neural developmental and/or neural psychiatric disorders (*Fromer et al., 2016*; *Purcell et al., 2014*). Evidence implicating particular genes in language impairments and autism is often established through genome-wide analysis such as screening for mutations and/or copy number variations in human subjects (*O'Roak et al., 2012*; *Sanders et al., 2012*; *Yuen et al., 2015*). Our findings showing that these genes are expressed in the basal ganglia, and that alterations in their expression are accompanied by impairments in vocal communication, provide additional evidence that these genes function in language processes and development. Among the genes we tested, *CNTNAP2* and *DISC1* have each been implicated in multiple disorders, including language impairments, ASDs, ADHD, intellectual disabilities, and schizophrenia (*Fromer et al., 2016*; *Purcell et al., 2014*), suggesting that these neural developmental disorders, cognitive impairments, and psychiatric disorders share common molecular substrates. It is possible that distinct but overlapping phenotypes are manifested depending on where and when these genes are expressed and how they are regulated, emphasizing the need to study these genes in the context of specific functional neural circuits.

miR-9 is expressed in the embryonic human brain and has been shown to regulate human *FOXP2* gene expression (*Fu et al., 2014*). It is likely that miR-9 plays a role in human language development by fine-tuning *FOXP1* and *FOXP2* expression, thereby coordinating the expression of a large number of genes that are active in neural development and function. Both *FOXP1* and *FOXP2* mRNAs have long 3'UTRs containing numerous miRNA binding sites, suggesting that these genes can be regulated by many miRNAs in addition to miR-9. Dysregulation of these miRNAs or of miRNA–*FOXP1/FOXP2* interactions by genetic, environmental, or physiological factors, thus, may contribute to language impairments and related neurodevelopmental disorders.

# Materials and methods

## Animals
Animal usage was approved by the Louisiana State University Health Sciences Center (LSUHSC) Institutional Animal Care and Use Committee. All experiments were conducted in male zebra finches (*Taeniopygia guttata*). Animals were housed in a 7 a.m. – 7 p.m. light-dark cycle. Juveniles at specific ages were obtained from our breeding colony at LSU School of Medicine, with each bird given an ID at hatching.

## Lentivirus production
The lentiviral vector used (a gift from Dr. M. Sheng) carries an mCherry fluorescent marker driven by the human ubiquitin C promoter (hUBC)(*Edbauer et al., 2010*). The zebra finch genome contains three genes encoding miR-9: *miR-9–1*, *miR-9–2*, and *miR-9–3* (*Luo et al., 2012*). We amplified a 300-nt genomic DNA fragment containing the *miR-9–3* precursor sequence from the zebra finch genome and inserted it downstream of mCherry in the lentiviral vector to generate the *miR-9*-expressing virus. The lenti-control virus is an empty vector. For lentivirus packaging and production, viral vectors and packaging plasmids were transfected into 293LTV cells (Cat. No. LTV-100, Cell Biolabs) using the calcium-phosphate method following the manufacturer's instructions (Clontech). Cell identity and the absence of mycoplasma contamination were confirmed by the vendor. Viral particles were harvested 48 and 72 hr after transfection. The crude supernatant was filtered through a 0.45 μm filter, spun at 2000 RPM, and collected and spun again by ultracentrifugation (25,000 rpm x 2 hr). The precipitated viral particles were resuspended in 50 μl PBS. Typically, we obtained virus suspensions with titers in the range of $1-5 \times 10^9$/ml.

## Stereotaxic injection
Juvenile birds were separated from their fathers at day 10 (10 d), and raised by their mother in a sound-proof chamber until 30 d. PCR was performed to determine the sex of juveniles (see Supplementary Table 2 for primer sequences). In assigning animals for viral injection, each animal had an equal probability of being injected with the control or miR-9 virus. Viral injection was performed on males at about 25 d. Stereotaxic injection was performed using a stereotaxic device including a head holder (Myneurolab) and a hydraulic microinjector (Narishige). The glass needles used for

injection have an inner tip diameter of 25 µm. The stereotaxic coordinates for injection into juvenile Area X were: anterior/posterior 2.8 and 3.2 mm, dorsal ventral 4.2 and 4.4 mm, and medial/lateral 1.3 and 1.5 mm using the bregma point as a reference. Animals were anesthetized with ketamine and xylazine. Each Area X received a viral injection at six or eight sites, 120–150 nl viral suspension per site. To facilitate diffusion of viral particles, the injection needle was allowed to remain at the site for 3–5 min before removal. For behavioral experiments, virus was injected bilaterally. For gene expression experiments, the lenti-miR-9 virus and the lenti-control virus were injected into Area X of opposite hemispheres.

To ensure that the impairments in vocal learning and performance that we observed were due to virally transduced miR-9 expression in Area X, and not due to Area X tissue damage caused by the injection process, we sacrificed the birds after the last song recording, and examined their Area X. All birds showed bilateral expression of virally transduced mCherry in Area X; the average area exhibiting strong mCherry signal accounted for 15–20% of total Area X volume. We also observed scattered cell bodies or dendrites showing an mCherry signal outside the core infected region but within Area X, suggesting that virally transduced gene expression spread beyond the core infected region. There was no difference in total Area X volume between the non-injected and injected groups, and there was no difference in total Area X volume between the lenti-control- and lenti-miR-9-injected groups (*Figure 1—figure supplement 2A*). In a separate experiment, we also quantified the number of neurons in juvenile Area X one month after viral injection using immunostaining of the neuronal marker Hu. We found similar numbers of Hu+ neurons in Area X of non-injected animals and in Area X injected with the lenti-control virus or with the lenti-miR-9 virus (*Figure 1—figure supplement 2B*). These results indicate that physical damage to Area X caused by viral injection, if any, was minimal; thus, the behavioral phenotypes that we observed are likely due to virally transduced miR-9 overexpression.

## Song behavior and analysis

By 30 days of age, the mother was removed and an adult male tutor was introduced to one miR-9-virus- or control-virus-injected juvenile (pupil). Both the pupil and the tutor were kept together in a sound-proof recording chamber until after 70 d. Songs of pupils were recorded at 65 d, 80 d, 100 d, and 150 d using a microphone (Technica AT803B), an eight-channel computer interface (M-Audio 2626), and SAP software version 1.02. Undirected songs were recorded automatically over two days for each bird at each age. Directed songs were recorded (from the same groups of birds) manually in the morning. Males were induced to sing directed songs by presenting female birds in a nearby cage; the females were changed every 10 min. Songs were classified as directed songs when the male sang facing the female as observed by an experimenter.

Zebra finches sing in bouts. A song bout typically contains multiple renditions of a motif, and a motif contains 5–8 distinct syllables that are rendered in a fixed sequence. A syllable is defined as a continuous segment of sound separated from another syllable by a silence gap, and each syllable can be quantitatively described by a set of distinct acoustic features using the software package Sound Analysis Pro (SAP). To select songs for analysis (for all song analyses described here unless otherwise stated), we manually sorted all song files recorded in one day from 8 a.m. to 12 p.m. and eliminated files representing cage noise. For each pupil, we typically selected 20 song files, approximately evenly spread across the entire set of song files (e.g., if there were 200 song files, the 1st, 11th, 21st, 31st, etc. were selected).

### Syllable omission analysis

We manually counted all syllables and syllable types in 20 song files (50–80 motif renditions) of a pupil and 10 song files (25–50 motif renditions) of its tutor. In cases in which a tutor or a pupil sang multiple versions of motifs, all versions were included in counting. Partial motifs typically appearing at the beginning or the end of a recorded song file were excluded from analysis.

### Motif similarity and syllable accuracy analysis

SAP first segments pupil and tutor song motifs into short (9 ms) segments, then performs pairwise comparisons between the short song segments along a motif pair and calculates a similarity score. For each pupil, we performed pairwise comparisons between 20 motif renditions by a pupil and 10

motif renditions by its tutor using the asymmetric mode of SAP (*Tchernichovski et al., 2000*). The resulting measurements of 200 pairwise comparisons were averaged to generate a motif similarity score. For data presented in *Figure 3A*, two investigators, one blinded to the treatment groups, performed similar analyses using two different sets of song files and motifs. The inter-observer reliability between the two analyses was 0.89. For maximum motif similarity, we ranked the 200 measurements and averaged the ten highest values (top 5%) to obtain the maximum motif similarity score. For syllable accuracy analysis, we measured the accuracy score for each syllable of a pupil's song motif in 20 renditions using the SAP. The accuracy scores of all syllables in a pupil's motif were averaged to generate a syllable accuracy score for that pupil.

## Acoustic feature analysis

The acoustic features (duration, mean frequency, goodness of pitch, frequency modulation, and Wiener entropy) of each syllable in a motif were measured using SAP. For pupils, 20 motif renditions were analyzed, and for tutors, 10 motif renditions were analyzed. For each syllable, the percentage difference from the tutor was obtained by subtracting a tutor's measurement from its pupil's measurement and then dividing by the tutor measurement. Then the percentage difference values of all syllable types of each bird were averaged. A coefficient of variation was calculated for each acoustic feature for each syllable in 20 renditions, and the coefficients of variation for all syllable types for each bird were averaged.

## Syllable transition entropy

We segmented songs recorded in two days from 8 a.m. to 12 p.m. using the auto-segmentation function of SAP. This provided us approximately 10,000–19,000 song syllables for each bird. Next, we used the clustering module of SAP to automatically classify these syllables into types (clusters). We visually confirmed that these syllable types matched with the sonograms. Obvious cases of false classification (e.g., due to segmentation inconsistency) were manually corrected. We next calculated the transition frequency between all pairs of syllable types, resulting in a matrix. Thus, for a song motif containing five syllable types (t = A, B, C, D, and E), we calculated syllable transition frequencies for A to A, A to B, A to C, A to D, A to E, and B to A, B to B, B to C,... and so on. Then, for each syllable type t (each row in the matrix), we calculated the relative transition probability $p_t$ (p=transition frequency between a syllable pair divided by the sum of transition frequencies of all syllable pairs in a row). We then computed transition entropy for each syllable type t: $Entropy_t = sum (p_t*log[p_t])$. Next, we computed a weighted transition entropy for each syllable type ($Entropy_{tw} = Entropy_t \times$ syllable weight), so as to give higher weight to the more frequent syllable types. A syllable weight was defined as the transition frequencies of a given syllable type (sum of a row in the matrix) divided by the sum of transition frequencies of all syllable types (sum of the entire matrix). Finally, the overall transition entropy for a song was given by the average of transition entropies of all its syllable types.

## Constant fundamental frequency analysis

Following previously established methods (*Kao and Brainard, 2006*; *Leblois et al., 2010*; *Murugan et al., 2013*), syllables containing a segment with a constant fundamental frequency (harmonic stacks) were included in the analysis. The constant fundamental frequency of the same set of syllables in both the DS and UDS contexts was measured using the SAP. Typically, 20–40 syllable renditions from 20 song files in each context were analyzed. For data presented in *Figure 5B*, two investigators, one blind to the treatment groups, performed similar analyses. The inter-observer reliability between the two analyses was 0.71.

## Amount of singing

To quantify the amount of singing for each bird, all song files recorded between 8 a.m. and 12 p.m. on two days were segmented using the batch mode of SAP. This process generated the total number of syllables a bird sang during the indicated time. The number of syllables produced by each bird during the 8 a.m. to 12 p.m. period in one day were then compared (*Figure 6—figure supplement 1*).

## Measuring virally infected regions in Area X and immunostaining Hu+ neurons

To evaluate the accuracy of injections into Area X and to measure the size of the virally infected region within Area X, after the last song recording, birds were euthanized and brains were sliced into 80 µm sagittal sections. Bright light and fluorescent images were scanned for each section using an Olympus BX61VS microscope equipped with VS-ASW FL software and a 2X lens. The size of Area X and the virally infected region (mCherry positive) within Area X was measured for all Area-X-containing sections using Image J software. For Hu staining, lenti-control and lenti-miR-9 viruses were injected into Area X of opposite hemispheres at about 30 d, and animals were euthanized one month later. Animals were anesthetized with ketamine and xylazine, followed by perfusion with PBS and fixation with 4% paraformaldehyde in PBS. Fixed brains were sliced into 30 µm sagittal sections. For each hemisphere, 3–4 sections containing mCherry signal within Area X were stained with an antibody against Hu (Cat#21271, Life Technologies; 1:500 dilution), followed by a fluorescein-conjugated goat anti-mouse secondary antibody (Cat#F2761, Invitrogen; 1:500 dilution). Images were taken with a Zeiss Axioplan2 fluorescent microscope (40X lens). For each section, the number of Hu + neurons (green fluorescence) in one or two microscope fields were counted using Image J. The experimenter was blind to treatment groups.

## Gene expression analysis

### Tissue collection

Four weeks after viral injection, animals were euthanized for gene expression analysis. All animals were observed in the morning for one hour (8–9 a.m.) before being euthanized, and no animal sang during this time. Animal brains were embedded in OCT medium and quickly frozen in dry ice. To obtain virally injected Area X tissue for gene expression (RNA or protein) analysis, we cryo-sectioned fresh frozen brains into 80 µm sagittal sections. Sections were first examined under a dissection microscope. Area X, a round structure less than 1 mm in diameter, visibly stands out from the surrounding areas. Sections containing Area X were then examined with a fluorescent microscope. Sections having an mCherry signal in Area X (confirmed by overlapping fluorescent and bright field images on a computer screen) were used for dissection. Sections were semi-fixed in 70% ETOH/PBS for 30 s, and a syringe needle (25G) was used to pick Area X tissue under a dissection scope, and to transfer it into a protein or RNA lysis buffer.

### Western blot analysis

Tissue was lysed with RIPA buffer containing protease inhibitor cocktail (Thermo, cat # 87786). Protein content was quantified with the BCA protein assay kit. Protein samples (20~25 µg) were separated on a 12% SDS-PAGE gel, and then transferred to PVDF membranes. The membranes were incubated overnight at 4°C with primary antibodies diluted in PBS-T containing 5% non-fat milk, followed by incubation with corresponding secondary antibodies for 2 hr at room temperature. Immunoreactive bands were detected using the ECL Chemiluminescence reagent. The following primary and secondary antibodies were used: anti-FOXP1 antibody (1:1000; HPA003876, Sigma-Aldrich); anti-FOXP2 antibody (1:1000; HPA000382, Sigma-Aldrich); anti-DARPP-32 antibody (1:3000; ab40801, Abcam); β-actin antibody (1:1000; sc-47778, Santa Cruz Biotech); goat anti-mouse IgG-HRP (1:2000; sc-2031, Santa Cruz Biotech); and goat anti-rabbit IgG-HRP (1:2000; sc-2030, Santa Cruz Biotech). Two bands with close molecular weights were detected for FoxP2 and for DARPP-32. As both FoxP2 and DARPP-32 can have posttranslational modifications, these bands likely represent posttranslational modification products of these proteins. We quantified the major bands with higher molecular weights. Quantification was normalized to β-actin. For peptide competition experiments, primary antibody was pre-incubated with respective competing peptide for FoxP2 (Cat. No. APrEST77852, Atlas Antibodies) or competing peptide for DARPP-32 (Cat. No. AB189245, ABCAM) following the manufacturer's guide before application to the blots. Protein extracted from Area X was used in these experiments. Concentrations of primary and secondary antibodies were the same as those described above.

## qRT-PCR

qRT-PCR was performed as described previously (*Shi et al., 2013*). Briefly, total RNA was isolated using TRIZOL reagent (Invitrogen). RNA was quantified using a Nanodrop spectrophotometer. To quantify miRNA expression levels, reverse transcription and qPCR were performed using the TaqMan microRNA assay kit following the manufacturer's protocol (Applied Biosystems). Briefly, reverse transcription was performed in a 15 μl reaction mix containing 10 ng total RNA, 3 μl miRNA primer mix, 1 mM dNTP, 50 U reverse transcriptase, and 3.8 U RNAse inhibitor. The PCR reaction was performed using the TaqMan probe mix in 10 μl TaqMan Universal PCR Master Mix. U6 small RNA was used as an internal control following the manufacture's recommendations. The specificities of the U6 small RNA and miRNA primers have been extensively tested and established by their manufacturer. To measure the expression of FoxP1 and FoxP2 and their downstream genes, reverse transcription was performed using 50 ng total RNA with the iScript Reverse Transcription Supermix kit (Bio-Rad). qPCR was performed using the iQ SYBR Green Supermix (Bio-Rad). *GAPDH* was used as a reference gene. Relative gene expression levels between experimental groups were determined using the comparative Ct ($2^{-\Delta\Delta Ct}$) method after normalizing to reference genes. For all samples, reverse transcription and qPCR were performed twice, and qPCR was carried out in triplicate. Primers for quantification of miR-9, miR-124, and U6 were obtained from Invitrogen (miR-9 — 4427975 and 000583; miR-124 — 4440886; and U6 — 4427975 and 001973). All other primers were obtained from IDT (Integrated DNA Technology); their sequences are listed in *Supplementary file 2*.

## Statistical analysis

All information related to statistical analysis is documented in the corresponding figure legends. Sample sizes were not statistically determined, but were similar to those generally employed in the field. In all figures (unless otherwise stated), each circle represents data from one animal. For data presented in *Figure 2*, *Figure 3A*, and *Figure 5B*, two investigators, one blind to the experimental groups, analyzed two different sets of song files. Two injected animals (one control pupil and one miR-9 pupil) were excluded from song behavior analysis because injection was outside of Area X. Data were assumed to have normal distributions, but this was not formally tested. Variance was assumed to be similar between groups, but this was not formally tested.

## Acknowledgements

We thank Dina Lipkind for advice on setting up recording chambers, Ashli Weber for helping sorting recorded song files, and Ellie Guillot for the art work in the figures. We also thank many members of the birdsong community for their constructive inputs through the course of this work.

## Additional information

### Funding

| Funder | Grant reference number | Author |
|---|---|---|
| National Institute of Mental Health | R01MH105519 | XiaoChing Li |
| National Science Foundation | 1258015 | XiaoChing Li |

The funders had no role in study design, data collection and interpretation, or the decision to submit the work for publication.

### Author contributions

Zhimin Shi, Data curation, Formal analysis, Investigation, Methodology; Zoe Piccus, Data curation, Formal analysis, Methodology; Xiaofang Zhang, Huidi Yang, Yan Ding, Data curation, Formal analysis; Hannah Jarrell, Data curation, Formal analysis, Validation; Zhaoqian Teng, Methodology; Ofer Tchernichovski, Software, Formal analysis, Methodology, Writing—review and editing; XiaoChing Li, Conceptualization, Formal analysis, Supervision, Funding acquisition, Investigation, Methodology, Writing—original draft, Project administration, Writing—review and editing

## Author ORCIDs

XiaoChing Li http://orcid.org/0000-0001-7544-494X

## Ethics

Animal experimentation: This study was performed in strict accordance with the recommendations in the Guide for the Care and Use of Laboratory Animals of the National Institutes of Health. All of the animals were handled according to approved institutional animal care and use committee (IACUC) protocol (#3187) of the LSU School of Medicine.

## Decision letter and Author response

Decision letter https://doi.org/10.7554/eLife.29087.030
Author response https://doi.org/10.7554/eLife.29087.031

# Additional files

## Supplementary files

• Supplementary file 1. Symbols, names, and functional descriptions of genes shown in *Figure 8*. Functional descriptions are based on NCBI (National Center for Biotechnology Information) database. For associations between genes and neural developmental and mental disorders, see the online database SFARI Genes (Simons Foundation Autism Research Initiative) and references therein. For genes not included in the SFARI Gene list (e.g., KCNA4 and SRPX2), see references cited in the main text.
DOI: https://doi.org/10.7554/eLife.29087.021

• Supplementary file 2. Sequences of primers used in all qRT-PCR experiments.
DOI: https://doi.org/10.7554/eLife.29087.022

• Audio file 1. A song file of tutor Blue 63 (the tutor for Gray 180 and Gray 139).
DOI: https://doi.org/10.7554/eLife.29087.023

• Audio file 2. A song file of control pupil Gray 180.
DOI: https://doi.org/10.7554/eLife.29087.024

• Audio file 3. A song file of miR-9 pupil Gray 139.
DOI: https://doi.org/10.7554/eLife.29087.025

• Audio file 4. A song file of tutor Black 45 (the tutor for Yellow 30).
DOI: https://doi.org/10.7554/eLife.29087.026

• Audio file 5. A song file of miR-9 pupil Yellow 30.
DOI: https://doi.org/10.7554/eLife.29087.027

• Transparent reporting form
DOI: https://doi.org/10.7554/eLife.29087.028

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
