## [Decision Letter]

Thank you for submitting your article "miR-9 regulates basal ganglia-dependent developmental vocal learning and adult vocal performance in songbirds" for consideration by *eLife*. Your article has been favorably evaluated by a Senior Editor and three reviewers, one of whom, Stephanie A White (Reviewer #1), is a member of our Board of Reviewing Editors.

The reviewers have discussed the reviews with one another and the Reviewing Editor has drafted this decision to help you prepare a revised submission.

Summary:

Shi et al. use the birdsong model to build upon their 2013 discovery that the language-related gene, FoxP2, is regulated by micro-RNAs within vocal control circuitry. The prior study established that miR-9 targets FoxP2. When birds song, miR-9 was shown to be up-regulated within Area X, the vocal dedicated subregion of the songbird basal ganglia, which resulted in FoxP2 down-regulation. In addition to being vocally regulated, this up-regulation depended upon the social context in which birds sang. To extend these observations, in the present study, the authors over-express miR-9 within Area X of young birds during the sensorimotor critical period for song development and examine the consequences on song learning, social-context dependent song variability in adults and on Area X protein and gene expression. A key finding is of broad song learning deficits. A second observation is that miR9 over-expression diminishes social context-dependent changes in song variability in adulthood. Finally, the authors show that these behavioral changes are accompanied by gene expression changes in Area X, including predicted FoxP2 targets and molecules involved in dopaminergic signaling, known to regulate social context variability. In sum, these studies provide a logical follow up to prior work and now provide a novel and compelling link between microRNA and vocal learning, with implications for human language learning.

Essential revisions:

1) Behavioral analyses: The crux of this study lies in altered song behavior. As such, the quality of the behavioral assays and the details of their description are critical.

1.1) Unusually high and consistent scores for learning in the control birds – This is epitomized by the statement: "Each control pupil imitated the song motif of its tutor completely – syllable by syllable in the same sequence" (given that no special tutoring paradigms appear to have been used). Moreover, Figure 2 recapitulate this '100%' learning and Table 1 shows 0 motif syntax changes across 6 pupils. In contrast, most studies reveal a range of learning in control animals. For example, the cited Haesler et al. study shows levels of learning in control birds ranging between 85 and 90%. Similarly in Heston et al., (2015) motif similarity ranged from 75-85%. If no variation can be shown in control birds it is a clear sign that the measurements are not sensitive enough to detect a normal range.

Given that the authors have collected all the song files, using an additional method to assess song is both feasible and warranted. That developed by Mandelblat-Cerf and Fee (2014) does not require users to segment motifs, and can thereby supply an alternative metric to substantiate and strengthen those used use here. A brand new option for assessing syllable types was just published in PLoS One. 2017 Jul 28;12(7):e0181992. doi: 10.1371/journal.pone.0181992. eCollection 2017. A fast and accurate zebra finch syllable detector. Pearre B, Perkins LN, Markowitz JE, Gardner TJ. These or another method of analysis should be used to corroborate the reported findings; alternatively, discrepancies should be reported.

1.2) The above issue relates to two methodological concerns: First, it is stated in the second paragraph of the subsection “Song recording and analysis” in the Materials and methods that motifs were 'randomly' selected (and later, birds were 'randomly' assigned to experimental groups). 'Random' has a precise scientific meaning and, unless e.g. some type of random number generator was linked to the selection of motifs, this explanation is insufficient. Second, it is stated that two experimenters evaluated song learning, only one of whom was double-blinded to the group assignment of the animal. Inter-relater reliability was then said to be 'similar'. The authors should provide metrics to substantiate this statement. Moreover, assessment of song variability as a function of social context should be performed blind to that context.

1.3) Experimental design: Were control and experimental pupils given the same tutor in order to control for tutor song complexity? This is important since, e.g. giving a simple tutor song to one group and a complex song to the other would skew the results regardless of molecular interventions. It appears that they were, as illustrated in Figure 2 where each type of pupil's song is compared to the same tutor song. This should be articulated.

1.4) Behavioral regulation of gene expression – A key consideration is the behavioral state of the animal at sacrifice. Prior work from this group, and from those of White and Scharff indicate that miR-9 and FoxP2 are behaviorally regulated, yet no mention is made of what the behavioral state of the animal was at sacrifice, nor of the time of day of sacrifice. As this could strongly influence gene expression data, this is a key issue to be explicit about.

1.5) Blocked learning versus developmental delay – Are the miR-9 pupils just developmentally delayed and could they eventually learn to produce a good copy of the tutor song? Figure 6 shows that many of these features improve with age in both miR-9 pupils and control pupils, so the poor imitation may merely reflect a developmental delay. Alternatively, poor imitation at 100d might reflect less practice (i.e., do the miR-9 pupils sing just as much as control pupils?). The authors could address this issue by quantifying the amount of singing in the two groups at the three different ages (65d, 85d, and 100d). In addition, they could analyze any songs of birds beyond 100d.

In the same vein, are the songs of miR-9 pupils crystallized? By 100d, are these birds singing a stereotyped sequence of song elements or are the songs of these birds highly variable from one song bout to the next? Previous studies have shown that lesions of Area X in juvenile birds prevent song crystallization, resulting in poor imitation (Introduction, second paragraph). In contrast, lesions of the downstream nucleus LMAN in juvenile birds results in premature crystallization and poor imitation (i.e., they only copy a few syllables of the tutor song). In both groups, the lesioned birds failed to produce an accurate copy of the tutor song but in completely different ways.

1.6) Interpretation – miR-9 overexpression from ~50d-100d could disrupt the ability of juvenile birds to 1) generate variable motor sequences; or 2) to select those motor programs that result in a good match to the tutor song model. Additional behavioral analyses would distinguish between these two possibilities:a) Quantify sequence consistency/linearity or transition entropy at different developmental stages. Does the variability in the sequence change over time?b) Instead of measuring the average similarity between the pupil's song and the tutor's song (Figure 2), quantify the maximum similarity score to determine whether miR-9 pupils are capable of producing good copies of the tutor song.c) Instead of measuring the difference in a particular acoustic feature between the pupil and student (Figure 3), quantify the "accuracy" of each syllable. Are miR-9 pupils able to produce accurate copies of the tutor song syllables? Or are the syllables more variable and also less accurate?

1.7) Data presentation – this relates primarily to Figure 3. The authors should clarify the y-axis in Figure 3. Is the absolute value of the difference between pupil and tutor plotted? Are all of the values really positive? Were any of values lower for the pupil's syllable versus the tutor's syllable? Figure 3 plots measures of particular features of a pupil's syllable and the corresponding tutor's syllables, but the number of syllables is not the same across groups (miR-9 v. control pupils) because control birds learned more syllables than miR-9 pupils. However, this makes direct comparison across groups difficult. It would help to indicate the shared syllables across the two groups to allow direct comparison. When comparing Figure 3 and Figure 3, it is not clear how there are significant differences in the goodness of pitch between control and miR-9 pupils when the values for all the syllables are completely overlapping. Again, here, it would help if one could evaluate the values for just the syllables that are matched across the groups.

1.8) Fundamental frequency analyses – Figure 5 – the fundamental frequency seems to be changing in the segment of the representative syllable (indicated by red and blue lines) that is plotted and analyzed. In this example, the frequency changes from a flat, constant frequency to a downsweep, raising some concern about the analysis. It would be useful to know that the inclusion of the last part of the syllable did not affect the measurement of the variance in the cFF across trials. While the segments used for control versus miR-9 pupils are probably the same across birds, variability in syllable duration was different across the two groups of pupils, and could potentially affect the measurement of cFF. The authors should confirm their measurements of cFF were restricted to constant frequency regions of harmonic stacks.

2) Molecular analyses:

2.1) Developmental versus experimental effects of miR-9 expression – the premise of overexpressing miR-9 makes sense in terms of trying to reduce FoxP2/FoxP1 expression. However, the authors have previously shown that miR-9 expression is developmentally regulated (upregulated). The authors should discuss the discrepancy between their previous finding that miR-9 levels increase from 45d-100d as song matures and becomes more similar to the tutor song (Shi et al., 2013) and the current finding that artificial overexpression of miR-9 in Area X results in greater variability in song (less mature) and lower similarity to the tutor song.

2.2) Cellular phenotypes affected by viral manipulation versus normal phenotypes that express miR-9 – is miR-9 expressed only in medium spiny neurons (MSNs) or is it expressed in other interneurons or the two types of pallidal neurons in Area X? The expression pattern of miR-9 in Area X is significant as it would suggest/constrain mechanisms by which miR-9 alters Area X function (e.g., striatal v. pallidal cell activity), and the authors should indicate which cell types express miR-9 (or cite a previous study).

2.3) Morphological effects – in addition, it would be useful to know whether there were gross changes in the structure of striatal MSNs or other neurons in miR-9 pupils as previous studies have shown that disrupted FOXP2 expression can affect spine density in MSNs in Area X.

2.4) Molecular targets – the focus on FoxP2/FoxP1 makes sense with previous work, but miR-9 has many other targets. How do the authors know that the "downstream" gene expression identified is really due to the change in FoxP2/FoxP1 expression? More detail would be helpful regarding the selected genes for analysis. Of course, it would have been possible to perform RNAseq to identify more genes that were differentially regulated between experimental groups. Moreover, beyond absolute gene expression levels, gene co-expression patterns reveal biologically relevant information. Those approaches may provide greater insight but do not negate the importance of the findings presented here. Rather, more detail regarding putative miR-9 binding sites in the selected genes, or whether the effects are thought to be mediated indirectly via FoxP2 would aid the interpretation of the results.

2.5) Protein analyses – Western blots should not be cropped and MW markers should be included. Given the many isoforms of FoxP1/2, known post-translational modifications, and antibody issues, this is important. Especially since multiple bands are sometimes observed as in Figure 7. Accordingly, 'For Western blots, bands with expected sizes were obtained' is overly vague and does not address the double bands present in Figure 7 for FoxP2 and Darpp32 respectively. Please clarify.

3) Statistics – subsection “miR-9 overexpression in Area X impairs adult song performance and abolishes social context-dependent modulation of song variability”, last paragraph: Different statistical tests were used for the same type of data; for 100d data, a paired t-test was used, but for 65d and 80d data of similar numbers, the non-parametric Mann-Whitney U test was used. The same tests should be used for the data at different ages or the authors should explain why different statistical tests were used. Given that normal distribution of the data is unlikely to be confirmed with these sample sizes, nonparametric tests and, ideally, bootstrapping or resampling methods are in order. The latter make no assumptions about the distribution of the data.

---

## [Author Response]

Essential revisions:1) Behavioral analyses: The crux of this study lies in altered song behavior. As such, the quality of the behavioral assays and the details of their description are critical.1.1) Unusually high and consistent scores for learning in the control birds – This is epitomized by the statement: "Each control pupil imitated the song motif of its tutor completely – syllable by syllable in the same sequence" (given that no special tutoring paradigms appear to have been used). Moreover, Figure 2 recapitulate this '100%' learning and Table 1 shows 0 motif syntax changes across 6 pupils. In contrast, most studies reveal a range of learning in control animals. For example, the cited Haesler et al. study shows levels of learning in control birds ranging between 85 and 90%. Similarly in Heston et al., (2015) motif similarity ranged from 75-85%. If no variation can be shown in control birds it is a clear sign that the measurements are not sensitive enough to detect a normal range.Given that the authors have collected all the song files, using an additional method to assess song is both feasible and warranted. That developed by Mandelblat-Cerf and Fee (2014) does not require users to segment motifs, and can thereby supply an alternative metric to substantiate and strengthen those used use here. A brand new option for assessing syllable types was just published in PLoS One. 2017 Jul 28;12(7):e0181992. doi: 10.1371/journal.pone.0181992. eCollection 2017. A fast and accurate zebra finch syllable detector. Pearre B, Perkins LN, Markowitz JE, Gardner TJ. These or another method of analysis should be used to corroborate the reported findings; alternatively, discrepancies should be reported.

Original Figure 2 may have caused some confusion. The data shown in original Figure 2/2D did not address motif similarity. Rather, they showed the numbers of copied syllable types and the average number of syllables per motif in control and miR-9 pupils. For example, if a tutor song had 7 syllables ABCDEFG, and the pupil song had the same 7 syllables, then this pupil was scored as coping 100% of the tutor syllable types. If the pupil song had syllables ABDEFG (missing syllable C), this pupil was scored as coping 85% (6/7) of tutor syllables. Indeed, no control birds omitted any tutor syllable types, but miR-9 pupils omitted some of the tutor’s syllables. To avoid this confusion, we have reorganized data presentation in Figure 2. The new Figure 2 shows the number of missing syllable types, and the new Figure 2 shows the average number of syllables per motif compared to the tutor in the control and miR-9 pupils.

Our motif similarity data are shown in Figure 3. Our control pupils have a motif similarity score of about 85-87%, consistent with the 85-90% and 75-85% ranges reported in the two cited papers (Haesler et al., 2007; Heston et al., 2015). SAP is widely used in quantifying zebra finch song learning, and it was used in several FoxP2 related papers including the two referenced above. Using the same software and similar methods in our analysis facilitates comparisons with those groups. In quantifying motif similarity, we compared 20 pupil motif renditions to 10 tutor motif renditions and averaged the 200 pairwise measurements for each animal. Two lab investigators (one blind to the treatment) analyzed two different sets of songs and obtained similar results (inter-observer reliability was 0.89). These, together with the new maximum similarity data (new Figure 3—figure supplement 1), syllable accuracy data (new Figure 3), and the similarity data for developmental groups (new Figure 5), helped substantiate the data presented in Figure 3. Thus, we did not perform additional similarity analysis using other analysis methods.

We analyzed syllable transition entropy in adult birds and showed the results in New Figure 4. We found that songs of miR-9 pupils have higher syllable transition entropy than songs of control pupils (also see our reply to point 1.6 below). Old Figure 2 and old Table 1 have been removed.

Regarding the statement “Each control pupil imitated the song motif of its tutor completely …”,

To avoid confusion, we revised the sentence to: “Control pupils imitated their tutors’ song motifs without syllable omission”.

1.2) The above issue relates to two methodological concerns: First, it is stated in the second paragraph of the subsection “Song recording and analysis” in the Materials and methods that motifs were 'randomly' selected (and later, birds were 'randomly' assigned to experimental groups). 'Random' has a precise scientific meaning and, unless e.g. some type of random number generator was linked to the selection of motifs, this explanation is insufficient. Second, it is stated that two experimenters evaluated song learning, only one of whom was double-blinded to the group assignment of the animal. Inter-relater reliability was then said to be 'similar'. The authors should provide metrics to substantiate this statement. Moreover, assessment of song variability as a function of social context should be performed blind to that context.

To select songs for analysis, we manually sorted all song files recorded from 8 a.m. to 12 p.m. and eliminated files representing cage noise. For each pupil, we selected 20 song files (10 for each tutor), approximately evenly spread across the entire set of remaining song files (e.g., if there were 200 song files, the first, 11th, 21th, 31th, … were selected). If a song file contained more than one motif rendition, we analyzed the one in the middle of the file, which is in general more complete. We have added a brief description of song selection to the Materials and methods (subsection “Song behavior and analysis”). For assigning juveniles for viral injection, we removed the word “random” and revised the expression to stipulate that each animal had an equal probability for being injected with the control or miR-9 virus (subsection 2 Stereotaxic injection”, first paragraph).

The motif similarity data were analyzed by two investigators: Zhimin Shi, who did most of the experiments described in this manuscript, thus was aware of the treatment groups. A student blind to the treatment groups did similar analysis using a different set of song files. The Inter-rater reliability between their data is 0.89 (subsection “Motif similarity and syllable accuracy analysis”).

Another lab member (who was blind to the treatment groups) analyzed cFF of UDS and DS during this revision. The new results are now shown in Figure 5. The inter-rater reliability between the new and previous results is 0.71 (subsection “Constant fundamental frequency analysis”).

1.3) Experimental design: Were control and experimental pupils given the same tutor in order to control for tutor song complexity? This is important since, e.g. giving a simple tutor song to one group and a complex song to the other would skew the results regardless of molecular interventions. It appears that they were, as illustrated in Figure 2 where each type of pupil's song is compared to the same tutor song. This should be articulated.

Because tutoring is a one-on-one lengthy process (~ 50 days), we used multiple tutors in the experiment, only a subset of pupils shared tutors (2 sets of triplets). For example, Grey180 (control pupil), Yellow23 (miR-9 pupil), and Grey139 (miR-9 pupil) shared a tutor. Their respective similarity scores were: Grey 180 = 83.8, Yellow23 = 68.4, and Grey139= 51.3; Yellow16 (control pupil), Yelow29 (control pupil) and Grey146 (miR-9 pupil) shared a tutor. Their individual scores were: Yellow16 = 87.6, Yellow29 = 82.5, and Grey146= 49.5. Although sample sizes are small, these data suggest that given the same tutor, control pupils learn better than miR-9 pupils. We were blind to the complexity of tutor songs when assigning tutors to pupils. If we use the number of syllables as an approximate indication of song complexity, the average numbers of tutor song syllables for the control and miR-9 groups are 6.2 and 5.7 respectively. Thus, it is unlikely that the apparent learning deficits of miR-9 pupils were a consequence of control pupils learning simpler tutor songs.

1.4) Behavioral regulation of gene expression – A key consideration is the behavioral state of the animal at sacrifice. Prior work from this group, and from those of White and Scharff indicate that miR-9 and FoxP2 are behaviorally regulated, yet no mention is made of what the behavioral state of the animal was at sacrifice, nor of the time of day of sacrifice. As this could strongly influence gene expression data, this is a key issue to be explicit about.

The behavioral state of the animals at sacrifice for gene expression experiments (both RNA and protein) was carefully controlled. All animals were observed in the morning for one hour (8 – 9 a.m.) before being sacrificed, and no animal sang during this time. This information is now provided in the Materials and methods subsection “Gene expression analysis.

1.5) Blocked learning versus developmental delay – Are the miR-9 pupils just developmentally delayed and could they eventually learn to produce a good copy of the tutor song? Figure 6 shows that many of these features improve with age in both miR-9 pupils and control pupils, so the poor imitation may merely reflect a developmental delay. Alternatively, poor imitation at 100d might reflect less practice (i.e., do the miR-9 pupils sing just as much as control pupils?). The authors could address this issue by quantifying the amount of singing in the two groups at the three different ages (65d, 85d, and 100d). In addition, they could analyze any songs of birds beyond 100d.In the same vein, are the songs of miR-9 pupils crystallized? By 100d, are these birds singing a stereotyped sequence of song elements or are the songs of these birds highly variable from one song bout to the next? Previous studies have shown that lesions of Area X in juvenile birds prevent song crystallization, resulting in poor imitation (Introduction, second paragraph). In contrast, lesions of the downstream nucleus LMAN in juvenile birds results in premature crystallization and poor imitation (i.e., they only copy a few syllables of the tutor song). In both groups, the lesioned birds failed to produce an accurate copy of the tutor song but in completely different ways.

We have now included analysis of motif similarity of 150d songs. These new data (subsection “The developmental process of vocal learning and performance”, first paragraph and new Figure 6) show that at 150d, the similarity score of miR-9 pupils was significantly lower than that control pupils (p = 0.008), and miR-9 pupils did not significantly improve their similarity score from 65d to 150d (p = 0.137). Figure 6 show that variations in acoustic features improved with age, but features of miR-9 pupils were more variable than those of control pupils for all ages.

We also quantified the amount of singing at 65d and 100d (subsection “The developmental process of vocal learning and performance”, last paragraph and Figure 6—figure supplement 1). We found that miR-9 pupils sang slightly more than control pupils at the two ages, but the differences were not significant (p = 0.419 for 65d and p = 0.109 for 100d groups). Thus, it is unlikely that the amount of singing is a determinative factor in impairments in song imitation in miR-9 pupils.

To address whether miR-9 pupils sing a stereotyped sequence of song syllables, we analyzed syllable transition entropy (also see our reply to point 1.6 below). We found that songs of miR-9 pupils have a higher syllable transition entropy (more variable syllable sequence) than controls (subsection “miR-9 overexpression in juvenile Area X impairs song performance and abolishes 158 social context-dependent modulation of song variability in adulthood”, first paragraph and new Figure 4). As pointed out by reviewers, previous lesion studies show that lesions in lMAN or Area X, two nuclei in the anterior forebrain pathway, have different consequences on song learning. While lesions in lMAN lead to prematurely stabilized song, birds with Area X lesions never achieve a fully stable song as control birds do (Scharff and Nottebohm, 1991). Although the experimental approaches used are very different (gene manipulation vs. electrolytic lesion), our observation is more aligned with the effects of lesions in Area X, suggesting that the effect of miR-9 overexpression (molecular manipulation) might be restrained by circuit functions, in this case functions of Area X (see Discussion, second paragraph for a brief discussion).

1.6) Interpretation – miR-9 overexpression from ~50d-100d could disrupt the ability of juvenile birds to 1) generate variable motor sequences; or 2) to select those motor programs that result in a good match to the tutor song model. Additional behavioral analyses would distinguish between these two possibilities:a) Quantify sequence consistency/linearity or transition entropy at different developmental stages. Does the variability in the sequence change over time?

We quantified syllable transition entropy for 100d songs. Indeed, syllable transition entropy of miR-9 pupils was higher than that of control pupils, which can be due to switching of syllable order, truncations of motifs and/or stuttering in miR-9 pupils (subsection “miR-9 overexpression in juvenile Area X impairs song performance and abolishes social context-dependent modulation of song variability in adulthood”, first paragraph, new Figure 4). In this new analysis, we included 10,000-19,000 auto-segmented syllables (recorded in the 8:00 a.m. to 12:00 p.m. period on two consecutive days) and used the clustering function of SAP to classify syllable types (see the Materials and methods subsection “Acoustic feature analysis”). Previously, we manually counted syllable sequence changes in 50-70 motif renditions and reported syllable sequence change in miR-9 pupils (old Table 1, now removed from the current manuscript). Although different methods were used, our new analysis and previous analysis point in the same direction: syllable sequence of miR-9 pupils is more variable.

However, we did not quantify syllable transition entropy at a younger age. Our preliminary analysis in younger birds showed that syllable classification errors during stages where motif units are not yet fully established would be high, but also difficult to detect. In other words, we suspect that in younger birds it would be too difficult to distinguish between misclassification due to spectral instability (which is higher in the miR-9 group) and cases of real variability in transitions across syllable types. Since syllables were less well defined or stable at a younger age, SAP often mistakenly assigned syllable/cluster IDs that could mislead the results by producing false high entropy scores.

b) Instead of measuring the average similarity between the pupil's song and the tutor's song (Figure 2), quantify the maximum similarity score to determine whether miR-9 pupils are capable of producing good copies of the tutor song.

We quantified the maximum motif similarity between the pupil’s song and the tutor’s song (subsection “miR-9 overexpression in juvenile Area X impairs vocal learning”, second paragraph and Figure 3—figure supplement 1). In similarity analysis, we compared 20 pupil renditions with 10 tutor renditions to generate 200 pairwise measurements. We ranked the 200 measurements, and averaged the 10 highest scores (top 5%) to represent the base score for each pupil. The maximum similarity of miR-9 pupils was significantly lower than that of control pupils (71% vs. 92%, p < 0.001), suggesting that at their best performance, miR-9 pupils were not able to produce good copies of the tutor song.

c) Instead of measuring the difference in a particular acoustic feature between the pupil and student (Figure 3), quantify the "accuracy" of each syllable. Are miR-9 pupils able to produce accurate copies of the tutor song syllables? Or are the syllables more variable and also less accurate?

We quantified syllable accuracy scores of control and miR-9 pupils, and found that miR9 pupils imitated their tutors’ syllables less accurately than did control pupils (subsection “miR-9 overexpression in juvenile Area X impairs vocal learning”, last paragraph, new Figure 3, and Materials and methods, subsection “Motif similarity and syllable accuracy analysis”).

1.7) Data presentation – this relates primarily to Figure 3. The authors should clarify the y-axis in Figure 3. Is the absolute value of the difference between pupil and tutor plotted? Are all of the values really positive? Were any of values lower for the pupil's syllable versus the tutor's syllable? Figure 3 plots measures of particular features of a pupil's syllable and the corresponding tutor's syllables, but the number of syllables is not the same across groups (miR-9 v. control pupils) because control birds learned more syllables than miR-9 pupils. However, this makes direct comparison across groups difficult. It would help to indicate the shared syllables across the two groups to allow direct comparison. When comparing Figure 3 and Figure 3, it is not clear how there are significant differences in the goodness of pitch between control and miR-9 pupils when the values for all the syllables are completely overlapping. Again, here, it would help if one could evaluate the values for just the syllables that are matched across the groups.

Old Figure 3 was confusing. The absolute values of the differences between pupil and tutor were plotted in old Figure 3. To be more clear, we have changed calculation methods such that the y-axis now reflects the direction of change (New Figure 3). The new results are slightly different from previous results because some positive and negative values canceled each other. In this new analysis, goodness of pitch is no longer significantly different between control and miR-9 pupils (p = 0.16). Thus, the PG panel together with the duration (not significant) panel have been removed. Because only a subset of pupils shared tutors, if we compared only matched syllables, many syllables would be excluded from the analysis. Thus, we measured features of all syllable types produced by a bird and averaged them to present that bird (see Materials and methods subsection “Song behavior and analysis”).

1.8) Fundamental frequency analyses – Figure 5 – the fundamental frequency seems to be changing in the segment of the representative syllable (indicated by red and blue lines) that is plotted and analyzed. In this example, the frequency changes from a flat, constant frequency to a downsweep, raising some concern about the analysis. It would be useful to know that the inclusion of the last part of the syllable did not affect the measurement of the variance in the cFF across trials. While the segments used for control versus miR-9 pupils are probably the same across birds, variability in syllable duration was different across the two groups of pupils, and could potentially affect the measurement of cFF. The authors should confirm their measurements of cFF were restricted to constant frequency regions of harmonic stacks.

We performed fundamental frequency analysis again by a person blind to the treatment groups. Our measurement was strictly restricted to the constant fundamental frequency part of harmonic stacks (not including the downsweep portion). The new results (New Figure 5) are consistent with the previous results (inter-observer reliability = 0.71). In our previous analysis, most measurements were restricted to the flat regions, but sometimes, the downsweep part may have been included. To reflect our new analysis, we replotted the blue line position in new Figure 5.

2) Molecular analyses:2.1) Developmental versus experimental effects of miR-9 expression – the premise of overexpressing miR-9 makes sense in terms of trying to reduce FoxP2/FoxP1 expression. However, the authors have previously shown that miR-9 expression is developmentally regulated (upregulated). The authors should discuss the discrepancy between their previous finding that miR-9 levels increase from 45d-100d as song matures and becomes more similar to the tutor song (Shi et al., 2013) and the current finding that artificial overexpression of miR-9 in Area X results in greater variability in song (less mature) and lower similarity to the tutor song.

We have previously shown that miR-9 expression in Area X increases from 45d to 100d during normal song development (Shi et al., 2013). In the present study, overexpression of miR-9 in juvenile Area X results in a song that is more variable and shows less similarity to the tutor song. A similar seemingly paradoxical observation was made for FoxP2: while in normal birds FoxP2 expression in Area X is higher in juveniles than in adults, knocking down FoxP2 in juveniles does not accelerate maturation, but results in a more variable song with lower similarity (Haelers et al., 2007; Murugan et al., 2013). It seems that miR9 overexpression (or FoxP2 knocking down) in juveniles impairs the progression of song maturation. A recent study shows that overexpression of FoxP2 in juvenile Area X caused inaccurate song imitation, raising the possibility that the dynamic regulation of behavioral-driven gene expression (i.e., FoxP2 or miR-9) plays a critical role in vocal development (Heston and White, 2015). Taken together, none of these gene manipulations (manipulating FoxP2 or miR-9) enhances song learning and maturation, an effect consistent with the earlier observation that lesions in juvenile Area X result in variable songs (Scharff and Nottebohm, 1991). Normal vocal development requires proper functioning of the entire circuit, which involves coordination among multiple song nuclei (e.g., HVC, lMAN, etc.). The effect of miR-9 overexpression (or FoxP2 knockdown) might be restrained by circuit functions, in this case, functions of Area X. Thus, changing gene expression in a manner opposite to their normal developmental pattern in Area X alone would not be sufficient to facilitate vocal learning (see Discussion, second paragraph).

2.2) Cellular phenotypes affected by viral manipulation versus normal phenotypes that express miR-9 – is miR-9 expressed only in medium spiny neurons (MSNs) or is it expressed in other interneurons or the two types of pallidal neurons in Area X? The expression pattern of miR-9 in Area X is significant as it would suggest/constrain mechanisms by which miR-9 alters Area X function (e.g., striatal v. pallidal cell activity), and the authors should indicate which cell types express miR-9 (or cite a previous study).

miR-9 is one of the most highly expressed miRNAs in vertebrate brains. By in situ hybridization and counter-staining with DAPI, we previously observed that miR-9 is expressed in almost all cells in normal Area X (Shi et al., 2013). The viral vector we used in the present study contains a ubiquitin promoter. Presumably, it could express miR-9 in most cell types in Area X. Area X contains multiple neuron types, including the predominant spiny neurons, pallidal-like neurons, and interneurons. Unlike mammals, where GPi and GPe neurons are segregated, in zebra finches, the pallidal neurons are intermingled with other neuron types in Area X. While GPi-like neurons project to the thalamic nucleus DLM, GPe-like neurons are less characterized.

FoxP2 is expressed in the spiny neurons (Haeslus et al., 2004); and given that FoxP1 is abundantly expressed in Area X, presumably it is expressed in spiny neurons as well (Teramitsu et al., 2004). It is not clear whether FoxP1 or FoxP2 is expressed in pallidal neurons in Area X, although in human brains, FOXP2 is detected in the GPi region (Teramitsu et al., 2004).DARPP-32, which functions downstream of dopamine receptors, is a well-established spiny neuron marker (Greengard et al., 1999). In zebra finches, dopamine signaling is found in both spiny and pallidal neurons (Leblois et al., 2010).Taken together, it is possible that the behavioral phenotype we observed resulted from a combination of miR-9 effects on spiny and pallidal neurons, but we cannot rule out possible contributions of other neuron types.

We agree with the reviewer that to fully understand functions of miR-9 in Area X, ultimately one needs to address its function in each specific neuron type. It will be important, ultimately, to use cell type-specific promoters to express miR-9 specifically and separately in spiny or pallidal neurons or interneurons to investigate effects of miR-9 on each neuron type.

2.3) Morphological effects – in addition, it would be useful to know whether there were gross changes in the structure of striatal MSNs or other neurons in miR-9 pupils as previous studies have shown that disrupted FOXP2 expression can affect spine density in MSNs in Area X.

Reducing FoxP2 in Area X affects spine density in spiny neurons in Area X (Schulz et al., 2010), and some of the genes we examined function in neurite growth and dendrite structural plasticity (Figure 8). Thus, it is likely that miR-9 overexpression affects dendrite and spine structure/density in spiny neurons in Area X. We agree with the reviewer, and also are eager to elucidate the morphological effects of miR-9. However, a systematic analysis of miR-9 effects on dendrite/spine structure of spiny and/or other neuron types would take considerable effort, and is beyond the scope of the current study. We anticipate investigating this issue in future studies.

2.4) Molecular targets – the focus on FoxP2/FoxP1 makes sense with previous work, but miR-9 has many other targets. How do the authors know that the "downstream" gene expression identified is really due to the change in FoxP2/FoxP1 expression? More detail would be helpful regarding the selected genes for analysis. Of course, it would have been possible to perform RNAseq to identify more genes that were differentially regulated between experimental groups. Moreover, beyond absolute gene expression levels, gene co-expression patterns reveal biologically relevant information. Those approaches may provide greater insight but do not negate the importance of the findings presented here. Rather, more detail regarding putative miR-9 binding sites in the selected genes, or whether the effects are thought to be mediated indirectly via FoxP2 would aid the interpretation of the results.

We selected FoxP1/FoxP2 downstream target genes based on published human and mouse studies, and we also gave preference to genes that have known neural functions and/or have been implicated in neural developmental/mental disorders. FOXP1/FOXP2 are known to bidirectionally regulate gene expression. Downregulation of FoxP1/FoxP2 by miR-9 could relieve repression, thus increasing gene expression. That 15 of the 26 genes we tested increased expression (Figure 8) is consistent with their being downstream from and being repressed by FOXP1/P2, and suggests that they are unlikely to be direct targets of miR-9. Other regulatory possibilities exist. For example, some downregulated genes potentially can be direct targets of miR-9, or are both downstream of FoxP1/P2 and targets of miR-9; some may be targets of other transcription factors that are regulated by FoxP1/FoxP2 or miR-9. It is also possible that one gene can be regulated by a combination of multiple mechanisms. Much work is needed to fully understand how each individual gene is regulated in this defined neural circuit.

miR-9 targeting sites in the 3’UTRs of FOXP1/FOXP2 are highly conserved between mammals and birds. But this may not be the case for other genes, since 3’UTR sequences are less conserved in general. The current databases of miRNA targeting sites (e.g., miRNA.org) do not include zebra finch 3’UTR sequences. The current zebra finch genome has numerous gaps, and many of the 3’UTR regions are not well annotated (e.g., we had to clone and sequence the FoxP2 3’UTR ourselves), making analysis of miR-9 target sites (and all other target sites) in the 3’UTRs of many zebra finch genes difficult.

RNA-seq is a powerful tool for unbiased analysis of a large number of genes. This approach may be explored in the future. However, RNA-seq is good for detecting robust differential gene expression. Gene expression changes we tested are moderate, ranging from 20-50%. We suspect we could have easily missed changes of this magnitude by using RNA-seq.

2.5) Protein analyses – Western blots should not be cropped and MW markers should be included. Given the many isoforms of FoxP1/2, known post-translational modifications, and antibody issues, this is important. Especially since multiple bands are sometimes observed as in Figure 7. Accordingly, 'For Western blots, bands with expected sizes were obtained' is overly vague and does not address the double bands present in Figure 7 for FoxP2 and Darpp32 respectively. Please clarify.

We replaced images in Figure 7/D with the original western blot images that contained samples of all four birds. We detected two FoxP2-related bands near 75kd from Area X. Two FoxP2-related bands with similar molecular weight also were detected from Area X by other groups using different anti-FOXP2 antibodies (Miller et al., 2008; Murugan et al., 2013; Heston and White, 2015). It is likely that the two bands represent isoforms or the post-translationally-modified FoxP2 proteins. We quantified the major band (the higher molecular weight band). We detected two DARPP-32-related bands near 32-35 kD from Area X. DARPP-32 has multiple phosphorylation sites. It is likely that the two bands represent posttranslational modified DARPP-32 proteins. We quantified the major band (the higher molecular weight band).

We also performed peptide competition experiments for FoxP2 and DARPP-32 using proteins extracted from Area X. We show that (Figure 7—figure supplement 1) preincubating primary antibodies with respective competing peptides (10-fold molar excess) inhibited the detection of the FoxP2- and DARPP-32-related bands.

3) Statistics – subsection “miR-9 overexpression in Area X impairs adult song performance and abolishes social context-dependent modulation of song variability”, last paragraph: Different statistical tests were used for the same type of data; for 100d data, a paired t-test was used, but for 65d and 80d data of similar numbers, the non-parametric Mann-Whitney U test was used. The same tests should be used for the data at different ages or the authors should explain why different statistical tests were used. Given that normal distribution of the data is unlikely to be confirmed with these sample sizes, nonparametric tests and, ideally, bootstrapping or resampling methods are in order. The latter make no assumptions about the distribution of the data.

Although Figure 5 both address cFF, data are presented differently. In Figure 5, cFF values of the same syllable in UDS and DS contexts were compared (the two dots connected by a line) for the 100d pupils. Thus, we used paired t-test for statistics. In Figure 5, the ratio of coefficients of variation in cFF of UDS over coefficients of variation in cFF of DS was compared between control and miR-9 pupils at 65d, 80d, and 100d (Figure 5 legend). We used Mann-Whitney test to compare data at all three ages. Consequently, the 100d data were evaluated twice: by paired t-test (Figure 5) at the single time point, and by Mann-Whitney U test (Figure 5) when included in multiple time points.